palaeontology/evolution

dinosaur, sauropodomorph, Triassic, muscle reconstruction, osteological correlates, locomotion

**Author for correspondence:**
Antonio Ballell
e-mail: ab17506@bristol.ac.uk

# Walking with early dinosaurs: appendicular myology of the Late Triassic sauropodomorph *Thecodontosaurus antiquus*

Antonio Ballell, Emily J. Rayfield and Michael J. Benton

School of Earth Sciences, University of Bristol, Life Sciences Building, Tyndall Avenue, Bristol BS8 1TQ, UK

 AB, 0000-0001-8901-2398; MJB, 0000-0002-4323-1824

Dinosaur evolution is marked by numerous independent shifts from bipedality to quadrupedality. Sauropodomorpha is one of the lineages that transitioned from small bipedal forms to graviportal quadrupeds, with an array of intermediate postural strategies evolving in non-sauropodan sauropodomorphs. This locomotor shift is reflected by multiple modifications of the appendicular skeleton, coupled with a drastic rearrangement of the limb musculature. Here, we describe the osteological correlates of appendicular muscle attachment of the Late Triassic sauropodomorph *Thecodontosaurus antiquus* from multiple well-preserved specimens and provide the first complete forelimb and hindlimb musculature reconstruction of an early-branching sauropodomorph. Comparisons with other sauropodomorphs and early dinosaurs reveal a unique combination of both plesiomorphic and derived musculoskeletal features. The diversity of appendicular osteological correlates among early dinosaurs and their relevance in muscle reconstruction are discussed. In line with previous evidence, aspects of the limb muscle arrangement, such as conspicuous correlates of lower limb extensors and flexors and low moment arms of hip extensors and flexors, suggest *Thecodontosaurus* was an agile biped. This reconstruction helps to elucidate the timing of important modifications of the appendicular musculature in the evolution of sauropodomorphs which facilitated the transition to quadrupedalism and contributed to their evolutionary success.

## 1. Introduction

The Triassic witnessed the origin and rise of the clade Dinosauria, a lineage of avemetatarsalian archosaurs that dominated land

ecosystems throughout the Mesozoic [1–3]. The first dinosaurs acquired a remarkably specialized skeletal morphology, including apomorphic modifications of the pelvic girdle and hindlimb [1,4,5]. These evolutionary innovations are thought to have had an impact on their posture and locomotion, suggesting a derived parasagittal and bipedal gait in the earliest members of Dinosauria [6]. While parasagittal gait had a deeper origin within Archosauria and bipedalism likely evolved several times in this clade [4,7–10], the early evolutionary success of dinosaurs has been largely attributed to their more efficient erect and bipedal locomotion [11]. The derived appendicular morphology and locomotion most likely played a key role in the evolution of Triassic dinosaurs and drastically shaped their biology and ecological roles within their ecosystems.

Sauropodomorpha is one of the main lineages of dinosaurs that had diversified by the Carnian [12–17], becoming ecologically diverse and globally widespread by the latest Triassic [18–20]. Early members of the clade were small to medium-sized gracile bipeds that later experienced a stepwise trend towards increased body sizes, eventually giving rise to the gigantic quadrupedal sauropods [21,22]. Reversion to a quadrupedal stance might have occurred in more than one sauropodomorph lineage, predicating the later evolution of gigantism [23]. Such locomotory shifts were promoted by multiple modifications of the appendicular skeleton from the plesiomorphic light-built limb architecture of early sauropodomorphs to the graviportal skeleton of sauropods [21,24], coupled with a drastic rearrangement of the locomotory musculature [25,26]. Characterizing the diversity of the locomotor apparatus in sauropodomorphs and the pattern of locomotor transformations it underwent is key to understanding the evolutionary history of the clade and the assembly of the unique sauropod body plan.

Myological reconstructions of fossil organisms are a valuable tool to inform functional inferences of different aspects of their palaeobiology [27,28]. The appendicular musculature of dinosaurs has been reconstructed based on osteological correlates of muscle attachment and comparisons to closely related extant taxa (e.g. [29–34]), and some accounts have focused on sauropodomorphs [26,35,36]. Identifying origin and insertion sites of girdle and limb muscles, and determining their lines of action, is fundamental to recreating the posture of non-avian dinosaurs and is the basis for biomechanical analyses of locomotion (e.g. [37–39]). Furthermore, the disparity of postural and locomotory modes in sauropodomorphs can be explored by describing the appendicular musculoskeletal anatomy of key taxa in a phylogenetic comparative context.

One such taxon is the early sauropodomorph *Thecodontosaurus antiquus* from the Late Triassic of Europe [40,41], which occupies a relevant phylogenetic position as the earliest branching post-Carnian sauropodomorph [42–46]. Numerous fossils of multiple individuals varying in body size and presumed ontogenetic stages are known, many of them showing beautifully preserved muscle scars and other osteological correlates [41]. The skeletal anatomy of *Thecodontosaurus* suggests it may have retained a plesiomorphic bipedal and cursorial locomotory mode [40,41,47], and preceded the origin of later-diverging members of the clade that acquired larger body sizes [22,23]. These reasons make *Thecodontosaurus* a valuable source of information about the appendicular muscular anatomy of early sauropodomorphs.

Here, we present a comprehensive reconstruction of the proximal appendicular musculature of *Thecodontosaurus antiquus* as the basis of functional inferences of its posture and locomotion. Comparisons between the musculoskeletal architecture of *Thecodontosaurus* and that of other early dinosaurs and sauropodomorphs are drawn to further document modifications to the pectoral and pelvic limb musculature throughout the early evolution of Sauropodomorpha, and their implications in the postural shifts that occurred within this lineage.

## 2. Material and methods

Our current understanding of *Thecodontosaurus* musculoskeletal anatomy is mostly based on material found in Tytherington, a Rhaetian fissure fill locality of southwestern England, UK [48]. This collection, housed at the University of Bristol Geology Department (BRSUG), is composed of hundreds of specimens of *Thecodontosaurus* of different sizes and ontogenetic stages [41]. References to *Thecodontosaurus* fossils from Durdham Down, a second locality in southwestern England [41], are occasionally made. The material preservation is remarkably good but varies between specimens due to the composition heterogeneity of the Tytherington fissure 2 breccia [48], as well as the preparation procedure. The best preserved specimens form the basis of the osteological correlates description (electronic supplementary material, table S1). Unfortunately, no complete articulated

specimen of *Thecodontosaurus* has been found, hindering the calculation of limb proportions and precise muscle orientations.

Inferences of muscle attachment sites are based on the extant phylogenetic bracket (EPB), which relies on the presence of osteological correlates associated with soft tissues of interest in extant bracketing taxa [28], in this case, crocodilians and birds. We also compare osteological correlates with lepidosaurs as the outgroup. Only muscles that are based on level I or level II inferences are reconstructed herein (table 1). A level I inference is supported by the presence of the muscle in both bracketing clades, while a level II indicates that the reconstructed muscle is found in only one of the lineages. More speculation is required in the case of muscle attachments that lack clear osteological correlates, which are indicated with the prime (') designation (table 1). In addition, this account only considers proximal limb muscles that have attachment sites on the girdles, stylopodial and zeugopodial elements of both fore- and hindlimb. This excludes the lower limb musculature, such as antebrachial, manus, shin, calf and pes muscles, and muscles connecting the pectoral girdle to the axial skeleton.

*Thecodontosaurus* fossils were carefully examined, looking for osteological correlates of muscle attachment, which include rugosities, smooth or scarred surfaces, ridges or fossae. The most conspicuous osteological correlates are usually related to muscles that attach via tendon or aponeurosis, leaving rugosities and scarring on the bone surface. Less evident are attachment sites that take the form of smooth surfaces, especially when muscles attach fleshly on the bone [49]. In the absence of clear osteological correlates, it may be possible to constrain the location of the muscle attachment on main bone structures or regions based on its position in extant bracketing taxa. An example is the deltopectoral crest, the site of insertion of different proximal forelimb muscles without evident osteological correlates in extant tetrapods. Despite the lack of muscle scars, the insertion of muscles like M. pectoralis can be confidently reconstructed on the deltopectoral crest. Both direct and indirect osteological correlates are considered herein to reconstruct muscle attachment sites. Direct correlates indicate the exact area of muscle attachment, such as rugosities and scars, while indirect correlates mark the boundaries of the attachment site, such as ridges or intermuscular lines [28,49].

# 3. Results

## 3.1. Pectoral and forelimb musculature

### 3.1.1. M. latissimus dorsi

This superficial muscle is sheet-like and has a characteristic fan-shaped morphology, with its insertion on the neural spines of the posterior cervical and anterior dorsal vertebrae [50,51] (figure 1). In birds, this muscle is divided in two heads, although a single muscle configuration is assumed to be the most likely condition in non-avian dinosaurs [26,32,34]. In extant archosaurs, the insertion of M. latissimus dorsi is on the proximal humerus, posterolateral to the deltopectoral crest, in the form of a proximodistal ridge, rugosity or other kinds of osteological correlates [32,51].

In *Thecodontosaurus*, M. latissimus dorsi most likely originated on the neural spines of the cervical and dorsal vertebrae, as inferred in other dinosaurs [26,34]. Otero [26] proposed the expanded dorsal tips of the neural spines or striations on these structures as osteological correlates of the muscle origin in sauropodomorphs. These features cannot be identified in *Thecodontosaurus* because neural spines are incomplete or abraded in cervical and anterior dorsal vertebrae. The insertion of M. latissimus dorsi on the humerus of *Thecodontosaurus* is represented by a proximodistally elongated rugose scar posterolateral to the deltopectoral crest (figure 2*a*,*c*). This muscle extends and abducts the humerus in crocodiles [51], while its main action in birds is humeral retraction [32]. M. latissimus dorsi would have retracted the humerus in *Thecodontosaurus*, similar to other bipedal non-avian dinosaurs [26,34].

### 3.1.2. M. pectoralis

M. pectoralis is the main ventral muscle of the pectoral girdle and is conserved across tetrapod lineages, sharing similar origin and insertion sites [52,53]. This fan-shaped muscle originates on the interclavicle, sternum and xiphisternum in lepidosaurs, and inserts on the deltopectoral crest [50,54]. In crocodilians, the ventral, parasagittal origin of M. pectoralis has fleshy attachment sites on the sternum, interclavicles and sternal ribs, and the muscle inserts on the medial side of the deltopectoral crest [51,55]. Similarly, the bulkier M. pectoralis of birds originates on the sternum and furcula and inserts on the medial surface of

**Table 1.** Reconstructed appendicular muscles in *Thecodontosaurus antiquus*. List of pectoral and pelvic muscles reconstructed, including origin and insertion sites and the respective levels of inference according to the Extant Phylogenetic Bracket. Levels of inference: I, unequivocal, the muscle is present in extant archosaurs; II: equivocal, the muscle is absent in either birds or crocodiles; I'/II', no osteological correlate present in *Thecodontosaurus*; I'/II'?, osteological correlate cannot be identified in *Thecodontosaurus* due to preservation.

| muscle | abbreviation | origin | level of inference | insertion | level of inference |
|---|---|---|---|---|---|
| **pectoral musculature** | | | | | |
| M. latissimus dorsi | mLD | neural spines of cervical and dorsal vertebrae | I? | scar on posterolateral side of humerus | I |
| M. pectoralis | mP | sternal plates and clavicles | I? | medial deltopectoral crest | I' |
| M. deltoideus clavicularis | mDC | lateral side of the acromion | I' | lateral side of deltopectoral crest | I' |
| M. deltoideus scapularis | mDS | posterolateral surface of scapular blade | I' | posterolateral surface of proximal humerus | I' |
| M. subscapularis | mSBS | medial side of scapular blade, on a depression with striations | I | medial tuber of humerus | I |
| M. subcoracoideus | mSBC | medial side of coracoid, anterior to coracoid pit | I' | medial tuber of humerus | I |
| M. supracoracoideus | mSC | dorsolateral surface of coracoid and lateral scapular fossa | I' | apex of deltopectoral crest | I? |
| M. coracobrachialis brevis | mCB | fossa on ventrolateral side of coracoid | I | anterior surface of proximal humerus | I' |
| M. scapulohumeralis anterior | mSHA | lateral surface of scapula posterodorsal to glenoid lip | I' | posterior surface of proximal humerus | I' |
| M. scapulohumeralis posterior | mSHP | ventromedial surface of scapular blade, ventral to the ventromedial ridge | I' | posterior surface of proximal humerus | I' |
| M. triceps brachii caput scapulare | mTBS | tuberosity on the lateral side of the glenoid lip of the scapula | I | olecranon process | I |
| M. triceps brachii caput lateralis | mTBL | posterolateral surface of the humeral shaft | I' | olecranon process | I |
| M. triceps brachii caput | mTBM | posteromedial surface of the humeral shaft | I' | olecranon process | I |
| M. biceps brachii | mBB | anterolateral surface of the coracoid | I? | anteromedial side of proximal ulna and radius | I' (ulna)/ I? (radius) |

*(Continued.)*

**Table 1.** (*Continued.*)

| muscle | abbreviation | origin | level of inference | insertion | level of inference |
|---|---|---|---|---|---|
| M. humeroradialis | mHR | fossa on lateral side of humerus | II | anteromedial side of proximal radius | II? |
| M. brachialis | mBR | cuboid fossa | I | anteromedial side of proximal ulna and radius | I' (ulna)/ I? (radius) |
| **pelvic musculature** | | | | | |
| M. iliotibialis 1 | mIT1 | dorsal margin of iliac preacetabular process | I | cnemial crest | I |
| M. iliotibialis 2 | mIT2 | dorsal margin of iliac body | I? | cnemial crest | I |
| M. iliotibialis 3 | mIT3 | dorsal margin of iliac postacetabular process | I | cnemial crest | I |
| M. ambiens | mAMB | pubic peduncle | I? | cnemial crest | I |
| M. femorotibialis medialis | mFMTM | anteromedial femoral shaft | I | cnemial crest | I |
| M. femorotibialis lateralis | mFMTL | anterolateral femoral shaft | I | cnemial crest | I |
| M. iliofibularis | mILFB | lateral surface of ilium, posterior to mif | I' | anterolateral rugosity on proximal fibula | I |
| M. iliofemoralis | mIF | lateral surface of ilium | I' | lesser trochanter | II |
| M. iliofemoralis internus | mIFI | preacetabular fossa of ilium | II | anteromedial proximal femur | I' |
| M. flexor tibialis externus | mFTE | lateral surface of postacetabular process | I | posteromedial surface of proximal tibia | I |
| M. adductor femoris 1 | mADD1 | Lateral surface of the proximal lamina of the ischium | I? | scar on posterolateral distal femur | I |
| M. adductor femoris 2 | mADD2 | groove on posterodorsal side of the ischial blade | II | scar on posterolateral distal femur | I |
| M. puboischiofemoralis externus | mPIFE | Pubis | II? | greater trochanter of femur | I? |
| M. ischiotrochantericus | mISTR | posterodorsal aspect of proximal ischium | II' | dorsolateral trochanter of the femur | I |
| M. caudofemoralis brevis | mCFB | brevis fossa of ilium | II | proximolateral surface of fourth trochanter | I |
| M. caudofemoralis longus | mCFL | anterior caudal vertebrae | I? | medial surface of fourth trochanter | I |

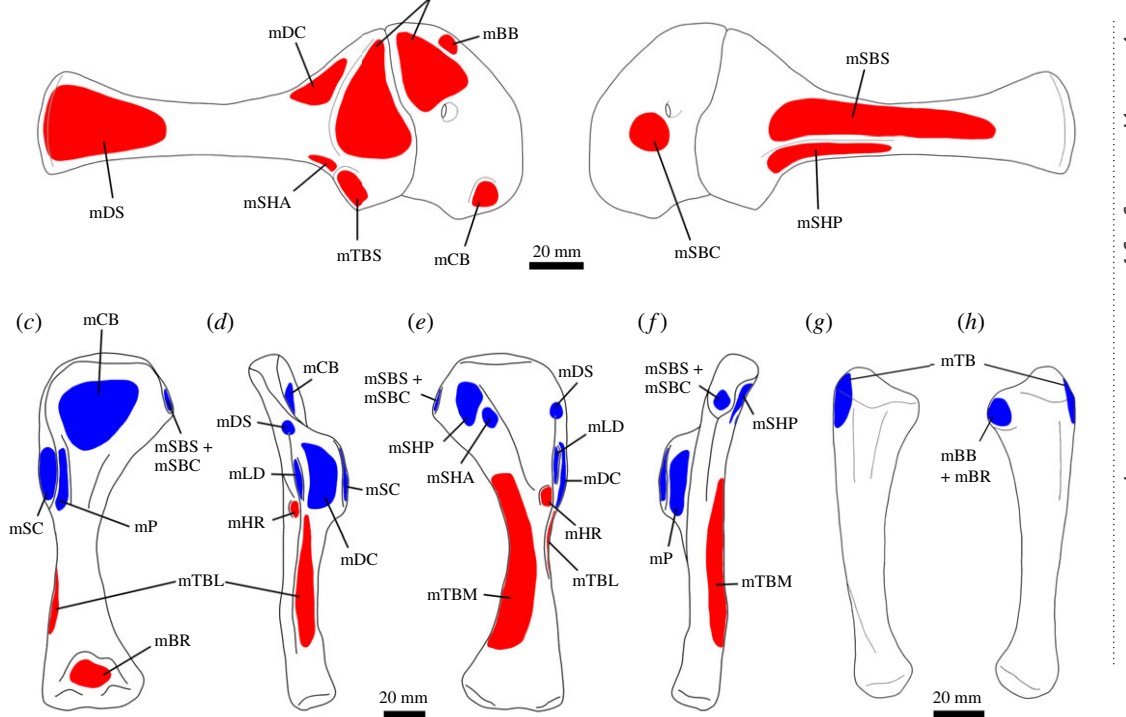

**Figure 1.** Pectoral appendicular musculature in *Thecodontosaurus antiquus*. Origin sites are shown in red and insertion sites in blue. Right scapulocoracoid in lateral (*a*) and medial (*b*) views. Right humerus in anterior (*c*), lateral (*d*), posterior (*e*) and medial (*f*) views. Right ulna in lateral (*g*) and medial (*h*) views. Abbreviations: mBB, M. biceps brachii; mBR, M. brachialis; mCB, coracobrachialis brevis; mDC, M. deltoideus clavicularis; mDS, M. deltoideus scapularis; mFU, M. flexor ulnaris; mHR, M. humeroradialis; mLD, M. latissimus dorsi; mP, M. pectoralis; mSBC, M. subcoracoideus; mSBS, M. subscapularis; mSC, M. supracoracoideus; mSHA, M. scapulohumeralis anterior; mSHP, M. scapulohumeralis posterior; mTBM, M. triceps brachii medialis; mTBL, M. triceps brachii lateralis; mTBS, M. triceps brachii caput scapularis.

the deltopectoral crest [32]. The number of subdivisions of M. pectoralis varies among and within these clades.

The arrangement of M. pectoralis in *Thecodontosaurus* is speculative because the elements from which it likely originated are missing. In early dinosaurs, these were the sternal plates, and possibly the clavicles [26,34], which are unknown to *Thecodontosaurus*. The insertion site is unequivocal, constrained by the similar location on the medial side of the deltopectoral crest (figure 1*f*) seen in extant archosaurs. While some early dinosaurs show muscle scars or tubercles near the apex of the deltopectoral crest [33,34], osteological correlates for the insertion of M. pectoralis are absent in *Thecodontosaurus*, as well as in other sauropodomorphs [26], suggesting a fleshy attachment [26]. M. pectoralis functions as the main humeral adductor.

### 3.1.3. M. deltoideus clavicularis

Inferences of M. deltoideus clavicularis origin and insertion sites are hindered by alternative homology hypotheses in birds [34,53,56]. In crocodiles, the fleshy origin of this muscle is on the anterior edge of the acromion [51,55], while it is located on the clavicle and interclavicle in lepidosaurs [50]. In both clades, the muscle inserts on the lateral surface of the deltopectoral crest of the humerus [50,51]. The most probable homologue in birds is M. propatagialis [56,57], which originates on the furcula and, in some taxa, also on the acromion process. The fleshy portion of M. propatagialis extends to the deltopectoral crest and inserts on the carpus via an elongated tendon or aponeurosis [32], a highly modified condition compared to other extant saurians.

The acromial process would have been the origin site of M. deltoideus clavicularis in *Thecodontosaurus*, similar to reconstructions of other basal dinosaurs [26,33,34]. The attachment area on the dorsolateral surface of the acromion is ventrally bounded by the acromial ridge and appears to show pitting (figure 3*a*), unlike other basal sauropodomorphs [26]. The insertion site is reconstructed

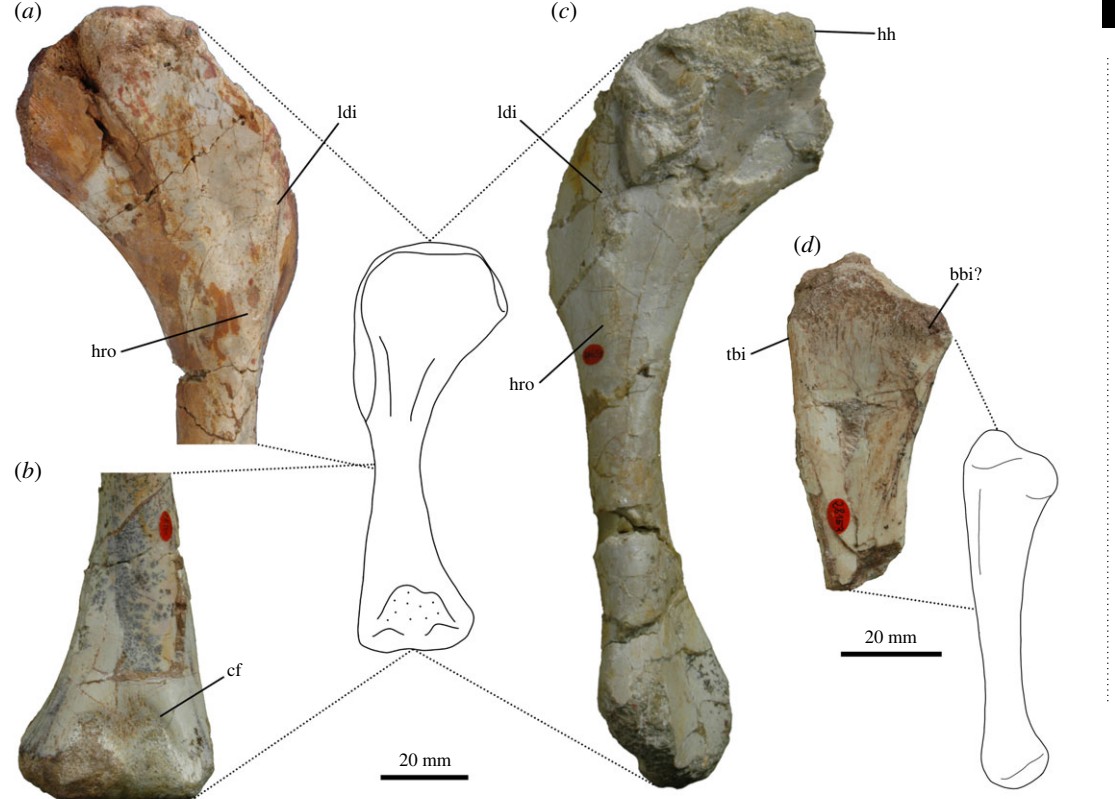

**Figure 2.** Osteological correlates of muscle attachment in the forelimb of *Thecodontosaurus antiquus*. Proximal portion of right humerus BRSUG 23608 in posterior view (*a*). Distal portion of left humerus BRSUG 28151 in anterior view (*b*). Left humerus BRSUG 26653 in posterior view (*c*). Proximal end of left ulna BRSUG 28153 in medial view (*d*). Abbreviations: cbo, M. coracobrachialis brevis origin; bbi, M. biceps brachii and M. brachialis insertion; cf, cuboid fossa; hro, M. humeroradialis origin; ldi, M. latissimus dorsi insertion; tbi, M. triceps brachii insertion.

occupying most of the lateral surface of the deltopectoral crest (figure 1*d*), the condition of crocodilians and lepidosaurs [50,51], although no muscle scar is present on this area, suggesting a possible fleshy insertion. With these attachment sites, M. deltoideus clavicularis would have contributed to humeral abduction and protraction [26,33,34].

### 3.1.4. M. deltoideus scapularis

In crocodilians and lepidosaurs, the origin of this muscle is located on the distal part of the lateral scapular blade, and it inserts tendinously on the posterolateral surface of the humeral head [50,51,55]. The homologue in birds, M. deltoideus major, is divided into two heads with variable origination sites, although towards a more anterior position relative to the origin in crocodilians. The posterior and larger head attaches to the acromion process and the furcula. The insertion site of M. deltoideus major varies among groups, usually present on the lateral side of the deltopectoral crest or along the humeral shaft [30,32].

In the absence of evident muscle scars or other osteological correlates in *Thecodontosaurus*, M. deltoideus scapularis is reconstructed here resembling the crocodilian configuration. The fleshy origin of this muscle does not leave a muscle scar on the scapular blade in extant saurians, a feature that is also absent in *Thecodontosaurus* and other basal dinosaurs [26,33,34], and thus the extent of this attachment along the scapular blade cannot be delimited with certainty. The insertion site would have been on the posterolateral humeral head proximal to the deltopectoral crest, similar to crocodiles [51], corresponding to the proximal end of the posterolateral ridge of the humerus. Other basal sauropodomorphs also lack a muscle scar on this area [26,36], although striations interpreted as the M. deltoideus clavicularis insertion are found in the basal theropod *Tawa* [34] and the ornithischians *Heterodontosaurus* and *Scutellosaurus* [33]. In *Thecodontosaurus*, this muscle would have acted as a humeral retractor and abductor [32,34], and might have contributed to long axis rotation [25,26,33].

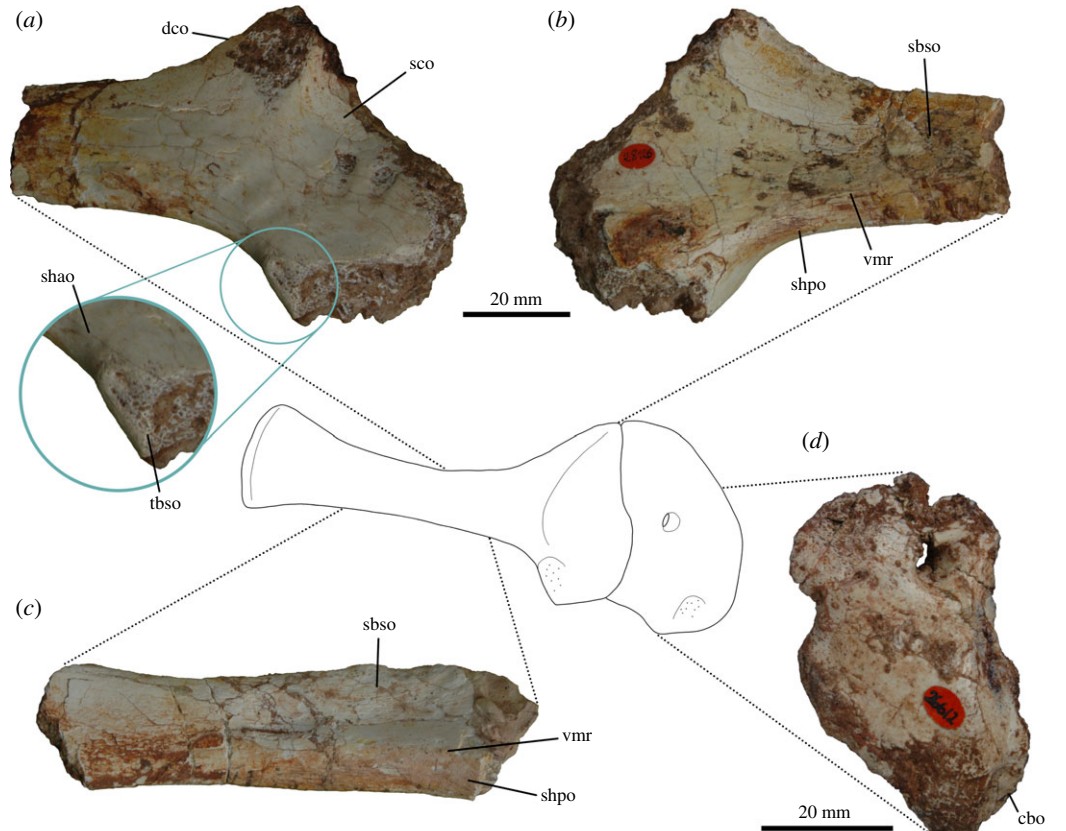

**Figure 3.** Osteological correlates of muscle attachment in the pectoral girdle of *Thecodontosaurus antiquus*. Anterior end of right scapula BRSUG 28126 in lateral (*a*) and medial (*b*) views. Left scapular blade BRSUG 29372–3495 in medial view (*c*). Right coracoid BRSUG 26612 in lateral view (*d*). Abbreviations: cbo, M. coracobrachialis brevis origin; dco, M. deltoideus clavicularis origin; sbso, M. subscapularis origin; sco, M. supracoracoideus origin; shao, M. scapulohumeralis anterior origin; shpo, M. scapulohumeralis posterior origin; tbso, M. triceps brachii caput scapularis origin; vmr, ventromedial ridge.

### 3.1.5. M. subscapularis

M. subscapularis represents one of the two divisions of the M. subcoracoscapularis present in lepidosaurs and birds [30,52]. In crocodilians, the two divisions remain undifferentiated, and the muscle is referred to as M. subscapularis [51,53]. M. subscapularis originates from the medial surface of the scapula in crocodilians [51,55], but its origin site is variable among lepidosaur species, present on the ventral or medial areas of the scapula and the suprascapula [30,54]. In most birds, M. subscapularis is divided into two heads originating from the medial and ventrolateral areas of the scapula [30,32]. In all these groups, M. subscapularis inserts via a tendon on the medial tuberosity of the proximal humerus [30,32,51,55].

In *Thecodontosaurus*, an elongated central depression on the medial surface of the scapular blade that bears striations represents the origin of M. subscapularis (figure 3*b,c*). Similar striations are present on the same area in *Saturnalia* [36]. The attachment area had a ventral boundary on the ventromedial ridge of the scapula, as suggested by previous muscular reconstructions in dinosaurs [26,34]. The insertion site of M. subscapularis was on the medial tuberosity of the humerus, although muscle scars cannot be identified because this structure is partially abraded in all *Thecodontosaurus* specimens. The insertion site would have been coincident with that of M. subcoracoideus. M. subscapularis would have had complex functions, operating as a humeral adductor and retractor and probably contributing to pronation and shoulder stabilization [25,26,32,34,51].

### 3.1.6. M. subcoracoideus

This subdivision of M. subcoracoscapularis is present as an independent muscle in lepidosaurs and birds, originating on the medial surface of the coracoid and sharing with M. subscapularis a tendinous insertion on the medial tuberosity of the humerus [32,54,58]. This general arrangement becomes more complex in some neognath and lepidosaur species that have two heads of this muscle [30,32].

The lack of M. subcoracoideus as an independent branch in crocodiles has been considered an autapomorphy of the clade [33], making the presence of a separate origination site unequivocal in dinosaurs. However, it is hard to identify the origination site on the medial side of the coracoid from lack of osteological correlates in *Thecodontosaurus* and other non-avian dinosaurs [26,30,34], as well as variability of this attachment in extant birds [32]. Burch [34] reconstructed the origin of the muscle in *Tawa* on the coracoid foramen, as seen in many neognaths [32], although other studies have suggested a more generalized position on the medial coracoid [26,33,59]. We reconstruct M. subcoracoideus in *Thecodontosaurus* originating in a variable position on the medial surface of the coracoid and inserting together with M. subscapularis on the medial tuberosity of the humerus. It would have adducted and rotated the humerus [32].

### 3.1.7. M. supracoracoideus

M. supracoracoideus constitutes a conserved portion of the shoulder musculature across tetrapods [53], although its architecture varies among clades. In lepidosaurs, this muscle originates from the lateral surface of the coracoid and coracoid cartilage and inserts on the deltopectoral crest or lateral tuberosity of the humerus via a tendon [30,50,54]. In crocodilians, M. supracoracoideus is divided into two or three heads originating from the medial and lateral surfaces of the scapulocoracoid and inserting on the apex of the deltopectoral crest [32,51,55]. This muscle is covered by M. pectoralis in birds, and its modified morphology makes it essential for flight as the main wing elevator [60]. M. supracoracoideus originates from the carina and sternum body, the clavicles and the coracoid, and its central tendon inserts on the lateral tuberosity of the humerus after passing through the triosseal canal, in a pulley-like configuration [32].

Reconstructions of M. supracoracoideus in extinct archosaurs are hindered by the high variation in attachment sites, number of subdivisions and muscle arrangement among bracketing clades and other tetrapod lineages. We follow previous reconstructions in non-avian dinosaurs [26,33,34,59] that consider M. supracoracoideus as originating on the lateral surface of the scapula and coracoid. This location is the site of origin of M. supracoracoideus intermedius and brevis in crocodilians [51]. Otero [26] argued that an origin on the medial side of the scapulocoracoid, similar to the crocodilian M. supracoracoideus longus, would create a peculiar arrangement of the muscle in sauropodomorphs due to the orientation of the pectoral girdle. Thus, M. supracoracoideus would have originated on the dorsal portion of the coracoid and scapula in *Thecodontosaurus*. The coracoid does not show clear osteological correlates of muscle attachment, but the scapular lateral fossa likely represents the origin of the muscle on this element. In *Thecodontosaurus*, the fossa is large and shallow, and it is delimited by the low acromial ridge (figure 3a), which represents the dorsodistal bound of the muscle. While multiple origin sites on the scapulocoracoid may have been present, as proposed by other reconstructions in sauropodomorphs [26], we consider a single muscle origin for simplicity, as in previous studies [25]. The insertion of the muscle is also ambiguous due to the different attachment sites in crocodilians and birds, although the configuration in the latter is considered derived, making the insertion on the tip of the deltopectoral crest the most likely in non-avian dinosaurs. In *Thecodontosaurus*, the tip of the deltopectoral crest is broken or abraded in all known specimens and thus the presence of muscle scars and the extent of the insertion of M. supracoracoideus cannot be determined. This muscle would have protracted and adducted the humerus, as in crocodilians [51] and presumably other early dinosaurs [25,26,33].

### 3.1.8. M. coracobrachialis brevis

M. coracobrachialis is generally present in extant tetrapods [53], although the number of subdivisions varies among clades. Lepidosaurs have two divisions, M. coracobrachialis brevis and longus, originating on the coracoid and inserting on the humerus [50,54]. Crocodilians only have the M. coracobrachialis brevis head, which is divided into ventralis and dorsalis heads [51,55], the latter likely being autapomorphic of this lineage [33]. In neognath birds, M. coracobrachialis also has two subdivisions, cranialis and caudalis [32]. The cranialis head is homologous to the M. coracobrachialis brevis ventralis of crocodilians [52], and while Dilkes [30] suggested that the caudalis head might have its homologue in the M. coracobrachialis longus of lepidosaurs, this is unlikely from the development and topology of both muscles [34,52,56]. In crocodilians, M. coracobrachialis brevis ventralis has a large fleshy origin on the lateral surface of the coracoid and inserts on the proximal humerus, medial to the deltopectoral crest [51,55]. M. coracobrachialis cranialis of birds originates on

the acrocoracoid process and inserts on the proximal humerus, distal to the humeral head and medial to the M. pectoralis insertion [32].

M. coracobrachialis brevis is reconstructed in *Thecodontosaurus* based on the homologous crocodilian ventralis head and avian cranialis head, as the presence of other subdivisions of M. coracobrachialis is equivocal in archosaurs. Based on these bracketing clades, the muscle has unequivocal origin and insertion sites in *Thecodontosaurus*, the former being on the ventrolateral side of the coracoid, most likely on an ovoid fossa located anterior to the glenoid (figure 3*d*). A similar osteological correlate is present in other sauropodomorphs [26], theropods [32,34] and ornithischians [33]. The insertion of M. coracobrachialis brevis is on the anterior surface of the proximal humerus, proximomedial to the deltopectoral crest. This surface shows a broad and shallow depression that tapers distally, as seen in other early dinosaurs [34,36], but lacks muscle scars. The position of the attachment sites indicates that M. coracobrachialis brevis had a main protractor activity, possibly including pronation and adduction [25,32–34].

### 3.1.9. M. scapulohumeralis anterior

This muscle is variably present in amniotes [53]. In lepidosaurs, M. scapulohumeralis anterior has its origin on the lateral side of the scapula around the scapulocoracoid fenestra and inserts on the posterior side of the proximal humerus, proximal to the M. latissimus dorsi insertion [50,54]. This muscle has been lost in crocodilians [61]. While absent in most palaeognaths and many neognaths, in birds that retain it M. scapulohumeralis anterior originates on the lateral surface of the scapula dorsal to the glenoid and inserts on the posterior side of the proximal humerus, in the fossa pneumotricipitalis [32].

Reconstructions of M. scapulohumeralis anterior in extinct archosaurs follow the fairly conserved configuration in extant birds and lepidosaurs. The origin of the muscle was on the ventrolateral side of the scapula dorsal to the glenoid. In *Thecodontosaurus*, the glenoid lip bears a prominent buttress for the attachment of M. triceps brachii caput scapulare, which is posterodorsally followed by a smooth and shallow concavity continuous with the scapular neck (figure 3*a*). This smooth concavity on the lateral side of the glenoid lip marks the origin of M. scapulohumeralis anterior. The posterior side of the proximal end of the humerus shows no muscle scar indicative of the insertion of this muscle in *Thecodontosaurus*, as in other non-avian dinosaurs [26,34]. However, its insertion would have been located on a shallow depression (posteromedial fossa in Ballell *et al.* [41]) distal to the humeral head and medial to the oblique ridge running along the posterior surface of the proximal half of the humerus (figure 2*a*). The attachment sites of M. scapulohumeralis anterior suggest it functioned mainly as a humeral retractor.

### 3.1.10. M. scapulohumeralis posterior

This muscle shows a similar arrangement in the two extant archosaur clades, being relatively more developed in birds. In crocodilians, it has its origin around the posterior edge of the scapular blade near the glenoid and inserts on the posterior side of the proximal end of the humerus [51]. In birds, its origin site covers most of the lateral surface of the scapula and the insertion site is in the crus ventrale fossae of the posterior side of the proximal humerus [32].

In *Thecodontosaurus*, M. scapulohumeralis posterior originates on the posterior edge of the scapula and extends onto the posteromedial surface of the scapular blade, which shows longitudinal striations. The ventromedial ridge is likely the boundary of this attachment site and separates it from the M. subscapularis origination area (figure 3*c*). The length of this ridge along the medial side of the scapular blade varies among sauropodomorphs, suggesting differences in the extent of the M. scapulohumeralis posterior origin [26]. The insertion site of the muscle would have been on the posteromedial fossa of the proximal humerus (figure 2*a*), proximal to the M. scapulohumeralis anterior insertion, although as with the latter, there is no clear osteological correlate for this attachment. As in modern birds and other bipedal non-avian dinosaur reconstructions [32–34], the role of this muscle in *Thecodontosaurus* would have been retracting and adducting the humerus—but see Otero *et al.* [25].

### 3.1.11. M. triceps brachii

The M. triceps brachii complex is present in all tetrapods [53], although the variable numbers of subdivisions in extant clades leads to complications in homologizing individual heads. In lepidosaurs,

this muscle complex is composed of three or four heads, one originating on the scapula and two on the humerus, and a variable coracoid head [50,54,58]. Among archosaurs, the number of heads varies from the five components of the crocodilian M. triceps brachii–a scapular, a coracoid and three humeral heads—to only three—a scapular, a vestigial coracoid and a humeral head—in birds [32,51]. In all these clades, the common area of insertion of all heads via a tendon is the olecranon process of the ulna [32,50,51,54].

M. triceps brachii caput scapulare (TBS), the scapular head of the complex, is present in lepidosaurs, crocodilians and birds (as M. scapulotriceps). In all groups, this division originates via a tendon on the posterolateral part of the scapula, dorsal to the glenoid lip, leaving a rugose scar [32,50,51,54].

The coracoid portion, M. triceps brachii caput coracoideum, is well-developed in crocodilians, tendinously attaching to posteriomedial areas of the coracoid and scapular blade [51], while in birds, M. coracotriceps is extremely reduced and vestigial [32].

The head arising from the medial side of the humeral shaft, M. triceps brachii caput mediale (TBM), is present in lepidosaurs, crocodilians and birds (M. humerotriceps). In lepidosaurs, the fleshy origin is located along the posteromedial surface of the shaft [50,54], while it occupies a relatively larger area on the posterior surface of the diaphysis in crocodilians and birds [32,51].

A second humeral portion, M. triceps brachii caput laterale (TBL), occupies a lateral position on the diaphysis in lepidosaurs and crocodilians. This portion originates from the posterolateral surface of the humeral shaft in lepidosaurs [54]. In crocodilians, this head has a linear origin on the lateral surface of the humeral shaft, along the base of the deltopectoral crest [51]. This head has been lost in birds [32].

The number of divisions of the M. triceps brachii complex in different clades of extinct archosaurs is unclear, although the scapular and medial heads are most likely present as in modern crocodilians and birds. Musculature reconstructions of dinosaurs also incorporate the lateral head because of its distinctive osteological correlate on the humerus [26,33,34], while the coracoid portion is seldom reconstructed [25,26]. Based on osteological correlates and the EPB approach, we infer the presence of M. triceps brachii caput scapulare, caput mediale and caput laterale in *Thecodontosaurus*. The coracoid head is not reconstructed in *Thecodontosaurus* because inference of its origin is hindered by the fragmentary nature of the coracoid, although this portion has been reconstructed in other early branching sauropodomorphs [25,26,36]. The origin of the scapular head on the glenoid lip is marked by an extensive rugose tuberosity (figure 3*a*) which is diagnostic of *Thecodontosaurus* [41] and indicates a tendinous attachment as in extant archosaurs. Muscle scars for the origins of the humeral heads of M. triceps brachii are not identifiable, although these likely occupied large portions of the posterior surface of the humeral shaft. An intermuscular line, distally continuous with the M. latissimus dorsi insertion, runs along the lateral side of the humeral shaft and probably separated the origination areas of M. triceps brachii caput medialis and caput lateralis. The medial head likely originated on most of the posterior surface of the diaphysis, posterior to the lateral intermuscular line. The origin of M. triceps brachii caput lateralis probably extended along the lateral side of the humeral shaft, anterior to the intermuscular line, either with a linear attachment similar to crocodilians [51] or a more extensive one as in lepidosaurs [54]. The three heads converged on a single tendon that inserted on the short olecranon process of the ulna. Its tip is abraded in all specimens, although the posterior surface of the ulna distal to the olecranon shows longitudinal striations that mark the distal part of the M. triceps brachii insertion (figure 2*d*). This muscle complex is the principal extensor of the forearm and also participates in humeral extension.

### 3.1.12. M. biceps brachii

M. biceps brachii is a conserved forelimb muscle with a consistent arrangement across tetrapods [52,53]. Among extant archosaurs, the configuration of M. biceps brachii is similar, originating on the anterior area of the coracoid and inserting via a tendon on the proximal ends of the radius and, in birds, of the ulna [32,51,55]. Crocodilians show a distinct muscle scar on the coracoid indicative of the long tendinous origin of the muscle [51]. In birds, the origin is marked by a facet on the anterolateral corner of the acrocoracoid, and the muscle has a second origination area on the crista bicipitalis of the proximal humerus [32,62].

Reconstruction of M. biceps brachii in *Thecodontosaurus* is somewhat speculative because the coracoid is incomplete, lacking its anterodorsal half where the muscle most likely originated. The coracoid tubercle, located on the lateral side of the coracoid anterior to the glenoid, is usually identified as the origin site of M. biceps in sauropodomorphs [36] and other dinosaurs [32,34]. An alternative origin

for this muscle has been proposed based on a ridge present on the anterodorsal portion of the lateral coracoid of derived sauropodomorphs. This possible osteological correlate resembles the ridge where the elongated origination tendon of M. biceps brachii attaches to the coracoid of crocodilians [26]. This 'biceps ridge' is not seen in earlier sauropodomorphs [36,46,63], phylogenetically closer to *Thecodontosaurus*, although its presence in this taxon cannot be determined. Thus, the origin of the muscle is likely on the anterolateral part of the coracoid, either anterior to the glenoid or more dorsally. Identification of the insertion site on the radius is hindered by poor preservation, and there is no sign of the distinct muscle scar present in several sauropodomorphs [26]. The ulnar insertion, inferred from its retention in lepidosaurs [50], birds [32] and probably crocodilians [64], was likely on the anteromedial process of the proximal end of the bone, which shows subtle striations (figure 2*d*). The main action of M. biceps brachii is flexing the elbow joint.

### 3.1.13. M. humeroradialis

This muscle is found only in *Sphenodon* [52] and crocodilians [51], and homology hypotheses for the muscle present in these two taxa and with the M. propatagialis of birds are poorly supported [56,57]. The crocodilian M. humeroradialis originates on the lateral side of the deltopectoral crest, distal to the M. deltoideus clavicularis insertion, leaving a distinct muscle scar. The tendinous insertion is also marked by an osteological correlate, a tubercle on the anterior side of the proximal end of the radius [51,55].

A level II inference is required to reconstruct M. humeroradialis in *Thecodontosaurus* and non-avian dinosaurs, unlike the level I of previously described forelimb muscles. Some non-avian dinosaurs show a tubercle in a similar position to that of the M. humeroradialis muscle scar in crocodilians [32]. *Thecodontosaurus* has a distinct oval fossa on this part of the humerus, distal to the M. latissimus dorsi insertion (figure 2*a*,*c*), and a similar feature is found in other basal sauropodomorphs [36,45,46], which might correlate with the origin of M. humeroradialis in these taxa. The insertion of the muscle in *Thecodontosaurus* is more speculative from the poor preservation of the radius, although it would have been located on the proximal end of the bone. With this arrangement, M. humeroradialis would have flexed the antebrachium.

### 3.1.14. M. brachialis

This anterior brachial muscle exhibits different attachment sites in extant archosaurs. Similar to the arrangement in lepidosaurs [50,54], the crocodilian M. brachialis takes its fleshy origin along the anterior surface of the humeral shaft distal to the apex of the deltopectoral crest and inserts via a shared tendon with M. biceps brachii on the proximal ends of the radius and ulna [51,64]. In birds, the shorter M. brachialis originates on the fossa musculus brachialis on the anterior side of the distal end of the humerus and inserts on the proximal end of the ulna [32].

Differing locations of the attachment sites of M. brachialis in extant archosaurs mean that reconstruction in extinct forms is conjectural. The origin of the muscle must be located along the anterior surface of the humeral shaft, from the apex of the deltopectoral crest to the distal end of the bone. However, early sauropodomorphs show a well-developed cuboid fossa on the anterior side of the distal humeral epiphysis between the distal condyles [36,46,65]. In *Thecodontosaurus* adults and juveniles, this fossa is deep and broad and has a profusely pitted surface (figure 2*b*), being a likely origination site for M. brachialis. This distal origin of the muscle is coincident with other dinosaur muscle reconstructions [26,36], while others place it proximally on the humeral shaft [25,34]. Its insertion is also speculative and might have been on the proximal ends of the zeugopodial elements, probably sharing a tendinous attachment with M. biceps brachii (figure 2*d*), as in crocodilians and lepidosaurs. M. brachialis was another forearm flexor in *Thecodontosaurus*.

## 3.2. Pelvic and hindlimb musculature

### 3.2.1. Mm. iliotibiales 1–3

The Mm. iliotibiales is a superficial muscle complex present across different tetrapod lineages [66–68] (figure 4). It has a sheet-like morphology, and the number of heads varies among clades, one in *Sphenodon*, two in squamates and three in archosaurs, all originating along the dorsal margin of the ilium [30,31,66]. In crocodilians and birds, the anterior portion of the complex, M. iliotibialis 1

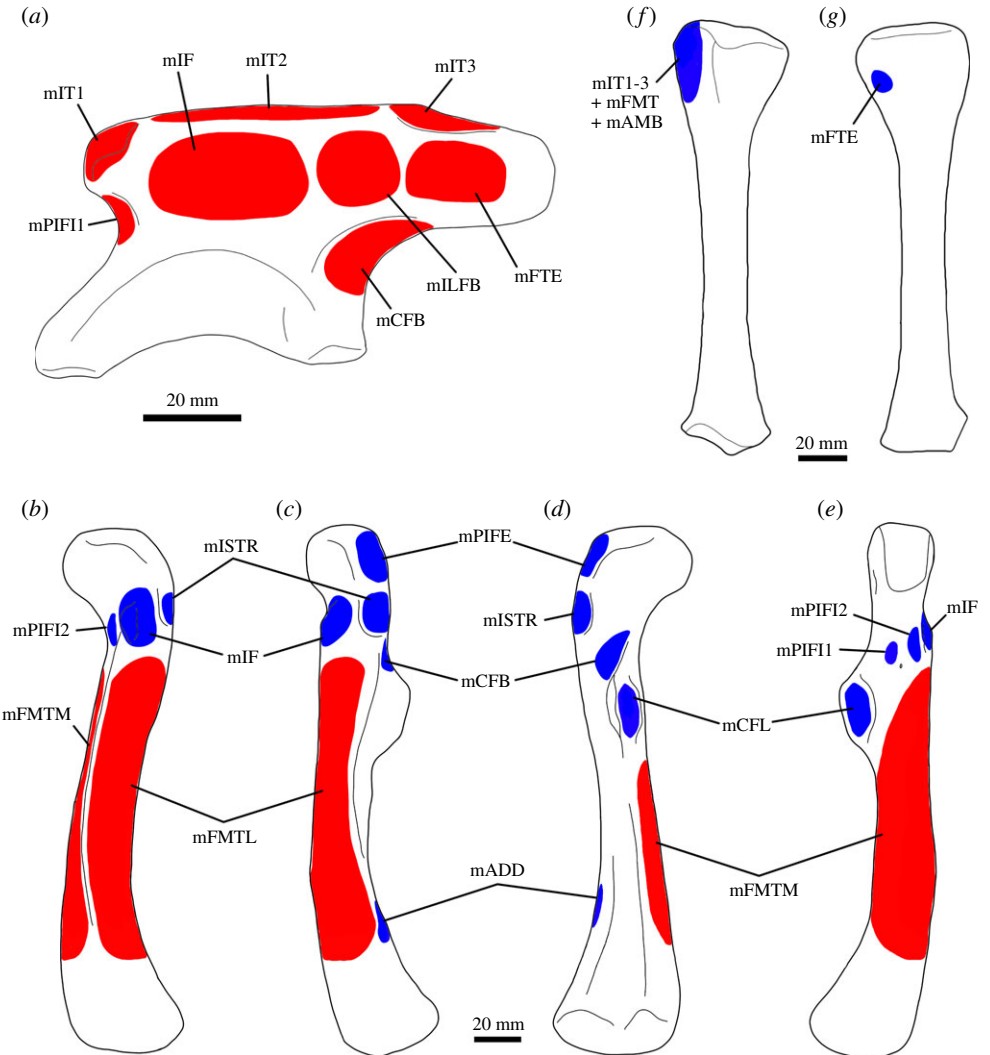

**Figure 4.** Pelvic appendicular musculature in *Thecodontosaurus antiquus*. Origin sites are shown in red and insertion sites in blue. Left ilium in lateral view (*a*). Left femur in anterior (*b*), lateral (*c*), posterior (*d*) and medial (*e*) views. Left tibia in lateral (*f*) and medial (*g*) views. Abbreviations: mADD, Mm. adductor femores; mAMB, M. ambiens; mCB, M. caudofemoralis brevis; mCL, M. caudofemoralis longus; mFMTM, M. femorotibialis medialis; mFMTL, M. femorotibialis lateralis; mFTE, M. flexor tibialis externus; mIF, M. ilifemoralis; mILFB, M. iliofibularis; mISTR, M. ischiotrochantericus; mIT1, M. iliotibialis 1; mIT2, M. iliotibialis 2; mIT3, M. iliotibialis 3; mPIFE, M. puboischiofemoralis externus; mPIFI1, M. puboischiofemoralis internus 1; mPIFI2, M. puboischiofemoralis internus 2.

(M. iliotibialis cranialis in birds), originates form the dorsal rim of the preacetabular process on a relatively small attachment site. M iliotibialis 2 is the largest head of the complex, originating posterior to M. iliotibialis 1 and from most of the dorsal margin of the ilium dorsal to the acetabulum. The origin of M. iliotibialis 3 is on the dorsal rim of the postacetabular process. In birds, M. iliotibialis 2 and 3 are collectively called M. iliotibialis lateralis and share an aponeurotic origin. All heads of Mm. iliotibiales converge into a common tendon that inserts on the cnemial crest of the tibia [30,31,55,62,66,69].

The presence of an Mm. iliotibiales divided into three heads is unequivocal in archosaurs and non-avian dinosaurs in particular. The preacetabular process of *Thecodontosaurus* has a striated surface (figure 5*a*,*b*), similar to the M. iliotibialis 1 scars in extant archosaurs [31], indicating the origin of the anterior head of the muscle complex. The dorsal margin of the ilium posterior to it, although poorly preserved in *Thecodontosaurus*, was the origination site of M. iliotibialis 2. This head would have been a broad and thin muscle sheet, covering the muscles of the deep dorsal group. The dorsal margin of the postacetabular process of *Thecodontosaurus* shows oblique striations lateral to the dorsal iliac crest (figure 5*a*,*b*), indicating the origin of M. iliotibialis 3. The three heads of Mm. iliotibiales likely

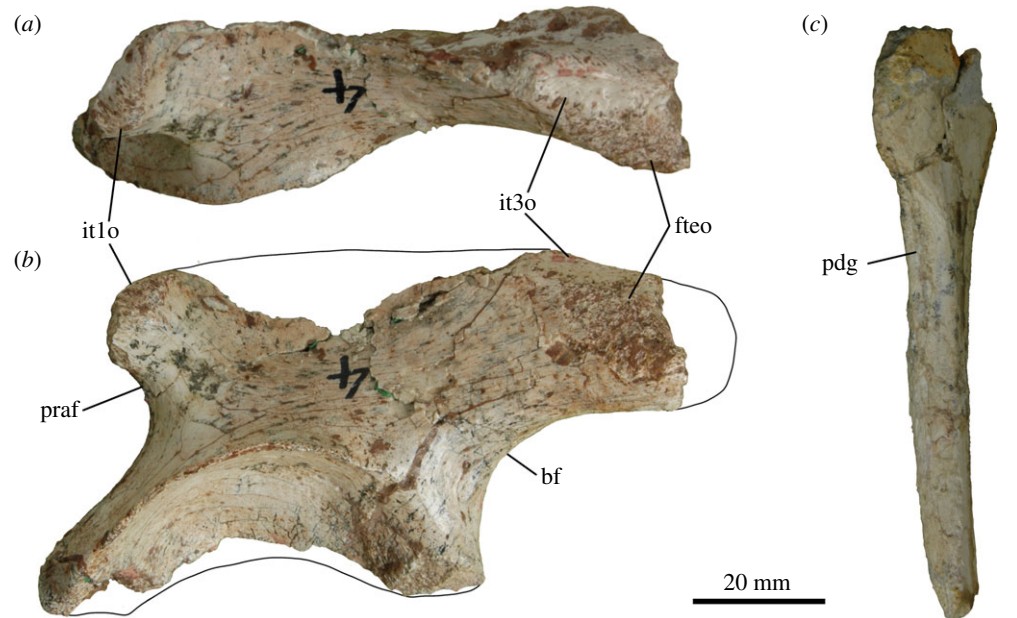

**Figure 5.** Osteological correlates of muscle attachment in the pelvic girdle of *Thecodontosaurus antiquus*. Left ilium BRSUG 23613 in dorsal (*a*) and lateral (*b*) views. Partial left ischium BRSUG 29372–3338 in posterodorsal view (*c*). Abbreviations: bf, brevis fossa; fteo; M. flexor tibialis externus origin; it1o, M. iliotibialis 1 origin; it3o, M. iliotibialis 3 origin; pdg, posterodorsal groove; praf, preacetabular fossa.

converged into the common extensor tendon together with M. ambiens and Mm. femorotibialis, inserting on the cnemial crest of the tibia, which shows a conspicuous area with striations (figure 6*g,h*). As in other bipedal dinosaurs, the Mm. iliotibiales complex would have primarily functioned as a knee extensor, while M. iliotibialis 1 flexes and M. iliotibialis 2 and 3 extend the hip joint [70].

### 3.2.2. M. ambiens

This superficial muscle is present across tetrapods, originating on the pubis and inserting on the tibia [68]. The M. ambiens of crocodilians is subdivided into two heads, both originating on the pubic peduncle and inserting on the cnemial crest via tendons [55,66]. Similarly, the origin of M. ambiens in birds is located on the preacetabular tubercle of the synsacrum [62,67].

The presence of M. ambiens in *Thecodontosaurus* is unequivocal, although its attachment sites are speculative because the pubis is not known. The muscle likely originated on the pubic tubercle of the pubis, as in other archosaurs [69], and inserted on the cnemial crest of the tibia (figure 6*g,h*) via a common tendon shared with Mm. iliotibiales and Mm. femorotibiales [31]. The main functions of M. ambiens are knee extension and hip flexion [70].

### 3.2.3. Mm. femorotibiales

Mm. femorotibiales is the third muscle complex of the Triceps femoris group, originating on the femoral shaft and inserting on the cnemial crest [66]. A single head is present in lepidosaurs, two in crocodilians and three in birds [50,62,66]. In crocodilians, M. femorotibialis internus is the largest head and its extensive origin on the anterior surface of the femoral diaphysis is separated from the smaller M. femorotibialis externus origin on the posterolateral side of the shaft by the M. iliofemoralis insertion [66,71]. The three heads of the muscle in birds are named M. femorotibialis medialis, intermedius and lateralis, and originate from the medial, anterolateral and lateral surfaces of the humeral shaft [62,72]. Mm. femorotibiales inserts on the cnemial crest via the extensor tendon shared with Mm. iliotibiales and M. ambiens [31].

Reconstruction of Mm. femorotibiales in dinosaurs requires a level I inference, although the number of heads is ambiguous. *Thecodontosaurus* femora show three distinct intermuscular lines along the diaphysis: the anterior intermuscular line, extending from the lesser trochanter to the anterior side of the lateral condyle; the posteromedial intermuscular line (= adductor ridge, Hutchinson [49]), running

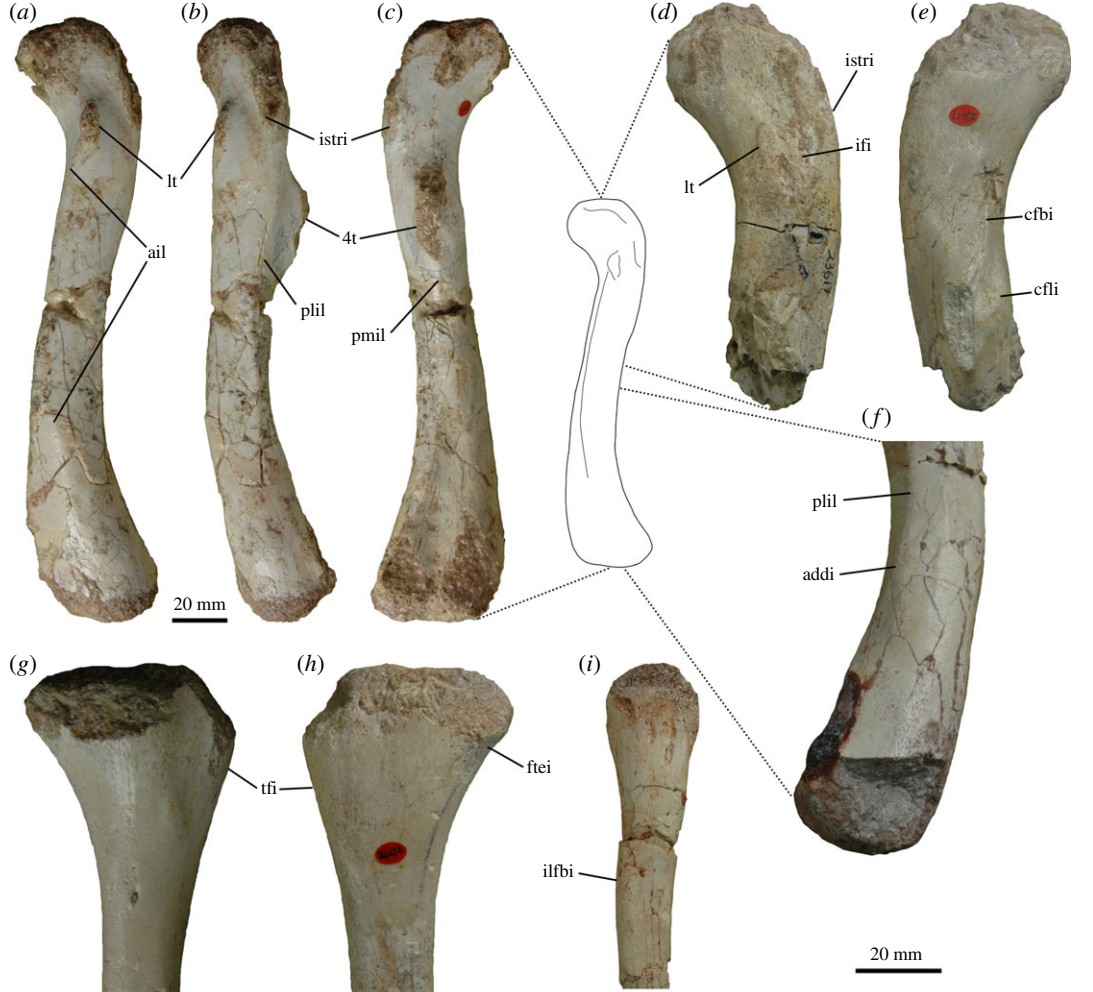

**Figure 6.** Osteological correlates of muscle attachment in the hindlimb of *Thecodontosaurus antiquus*. Left femur BRSUG 23615 in anterior (*a*), lateral (*b*) and posterior (*c*) views. Proximal portion of left femur BRSUG 23617 in anterior (*d*) and posterior (*e*) views. Distal portion of right femur BRSUG 29372-3311 in lateral view (*f*). Proximal portion of right tibia BRSUG 26656 in lateral (*g*) and medial (*h*) views. Proximal portion of left fibula BRSUG 26634 in lateral view (*i*). Abbreviations: 4t, fourth trochanter; addi, Mm. adductor femores insertion; ail, anterior intermuscular line; cfbi, M. caudofemoralis brevis insertion; cfli, M. caudofemoralis longus insertion; ftei, M. flexor tibialis externus insertion; ifi, M. iliofemoralis insertion; ilfbi, M. iliofibularis insertion; istri, M. ischiotrochantericus insertion; lt, lesser trochanter; pmil, posteromedial intermuscular line; plil, posterolateral intermuscular line; tfi, Triceps femoris insertion.

from the distal end of the fourth trochanter to the posterior side of the medial condyle; and the posterolateral intermuscular line (= posterior intermuscular line, Hutchinson [49]), connecting the greater trochanter and the posterior side of the lateral condyle [41]. These lines delimit the origin of the different heads of Mm. femorotibiales in archosaurs, as well as boundaries between the attachment sites of this complex and other hindlimb muscles [49]. In *Thecodontosaurus*, the anterior and posteromedial intermuscular lines marked the limits of the M. femorotibialis medialis origin, while that of M. femorotibialis lateralis was bounded by the anterior and posterolateral intermuscular lines (figure 6*a–c*). Both heads of Mm. femorotibiales joined the extensor tendon, inserting on the cnemial crest (figure 6*g,h*). This muscle complex would have acted as a knee extensor [70].

### 3.2.4. M. iliofibularis

M. iliofibularis is a component of the superficial dorsal group and is present across tetrapods [30,66,68]. In extant archosaurs, this muscle has its origin on the lateral surface of the ilium, posterior to the M. iliofemoralis origin and ventral to the posterior portion of the Mm. iliotibiales origin, and its insertion site is on the lateral side of the proximal fibula [30,31,62,66,73].

M. iliofibularis was unequivocally present in non-avian dinosaurs. The lateral surface of the iliac blade of *Thecodontosaurus* lacks a distinct osteological correlate for the origin of M. iliofibularis. Other dinosaurs show a distinct concavity on the posterior half of the iliac blade, anterior to the postacetabular process, interpreted as the origin of M. iliofibularis [31,33,74]. Despite the absence of such a trait in *Thecodontosaurus*, the origination site can be constrained to a similar position on the iliac blade, anterior to the M. flexor tibialis externus origin scar, ventral to the Mm. iliotibiales origin and posterior to the M. iliofemoralis origin. The insertion of the muscle is marked by a faint rugosity on the anterolateral surface of the proximal third of the fibula (figure 6*i*), a similar position to that of extant archosaurs and other non-avian dinosaurs [31,33,35]. M. iliofibularis would have acted as both hip and knee extensor in a erect biped like *Thecodontosaurus* [70].

### 3.2.5. M. iliofemoralis

M. iliofemoralis is a component of the deep dorsal group of tetrapods [68]. In archosaurs, the homologies among the derivatives of this muscle are well identified: the M. iliofemoralis of crocodilians corresponds to the M. iliofemoralis externus and M. iliotrochantericus caudalis of birds [66,75]. The crocodilian M. iliofemoralis originates on the iliac blade, dorsal to the acetabulum, and inserts along the lateral surface of the humeral shaft, between the M. femorotibialis internus and externus origination sites [66]. In birds, M. iliofemoralis externus originates on the ilium, dorsal to the acetabulum, posterior to the M. iliotrochantericus caudalis origin and anterior to the M. iliofibularis origin and inserts on a ridge on the proximolateral femur homologous to the trochanteric shelf [31,49]. The avian M. iliotrochantericus caudalis originates from most of the lateral surface of the preacetabular ilium and inserts on a part of the trochanteric shelf homologous to the lesser trochanter [49,72].

Reconstruction of the M. iliofemoralis musculature in non-avian dinosaurs is a level I inference, although its division into M. iliofemoralis externus and M. iliotrochantericus caudalis is equivocal. The ilium of *Thecodontosaurus* lacks any osteological correlate that indicates the division of this muscle mass, as in most non-avian dinosaurs [69]. In addition to this, the plesiomorphic trochanteric shelf of dinosauromorphs, retained by the basalmost sauropodomorphs [12,16,76,77], is lost in *Thecodontosaurus* and more derived members of the clade [41]. The presence of the trochanteric shelf has been interpreted as a possible indication of the differentiation of M. iliofemoralis into M. iliofemoralis externus and M. iliotrochantericus caudalis [49,78], so the loss of this osteological feature may represent a reversal to a single muscle in post-Carnian sauropodomorphs. Thus, we reconstruct an undivided M. iliofemoralis in *Thecodontosaurus*, originating on the central area of the iliac blade (figure 5*b*), dorsal to the acetabulum and anterior to the M. iliofibularis origin, and inserting on the lesser trochanter of the femur. The lesser trochanter and its surrounding area show conspicuous pitting in *Thecodontosaurus* (figure 6*d*), correlating with the insertion of the muscle. The action of M. iliofemoralis changed through the evolution of Dinosauromorpha, although a primarily strut-like abductor function has been suggested in early dinosaurs like *Thecodontosaurus* [78].

### 3.2.6. M. puboischiofemoralis internus 1

The homologies of the internal iliofemoral musculature of archosaurs were identified by Romer [66,68] and Rowe [75], who proposed that the M. puboischiofemoralis internus 1 of crocodilians is homologous with the M. iliofemoralis internus of Aves. M. puboischiofemoralis internus 1 originates on the ventromedial surface of the ilium and proximomedial ischium of crocodilians and inserts medial to the fourth trochanter [55,66]. The avian M. iliofemoralis internus originates on the preacetabular fossa of the ilium and inserts on a scar on the medial side of the proximal femur [49,62,69].

The presence of M. puboischiofemoralis internus 1 in non-avian dinosaurs is unequivocal, but there is uncertainty about its areas of attachment. Based on the attachment sites in extant archosaurs, the origin of the muscle could be located on the medial side of the ilium or on the preacetabular fossa. A shallow fossa with a pitted surface is located ventral to the preacetabular process of *Thecodontosaurus* (figure 5*b*), indicating that M. puboischiofemoralis internus 1 might have had a more lateral origin similar to that of birds, also presumed in other non-avian dinosaurs [31,33,69]. The insertion site is more equivocal because osteological correlates are absent on the anteromedial surface of the proximal femur, although the EPB suggests an attachment between the lesser trochanter and the fourth trochanter. With this arrangement, M. puboischiofemoralis internus 1 would have protracted the femur in *Thecodontosaurus* [39,70].

### 3.2.7. M. puboischiofemoralis internus 2

The second part of the M. puboischiofemoralis internus of Crocodylia and other diapsids has been considered homologous to the avian M. iliotrochantericus cranialis and medius [75]. In crocodilians, M. puboischiofemoralis internus 2 originates on the centra of the six lumbar vertebrae and inserts on the anterolateral side of the proximal femur, lateral to the M. puboischiofemoralis internus 1 insertion [66]. The anterior and middle heads of the M. iliotrochantericus of Aves originate on the ventral margin of the preacetabular ilium, ventral to the origin of the posterior head and anterior to the M. iliofemoralis internus origin, and insert on the trochanteric crest [49,62,72].

Reconstruction of a M. puboischiofemoralis internus 2 in non-avian dinosaurs is a level I inference, assuming homology with the avian M. iliotrochantericus cranialis and medius (but see [31]), although its subdivision and attachment sites are equivocal. The location of the origination site is particularly ambiguous, and muscle reconstructions in non-avian dinosaurs have placed it either on the posterior dorsal vertebrae [35,74] or the preacetabular process [31,79]. The reduced size of the preacetabular process in *Thecodontosaurus* and the lack of a clear muscle scar on the ilium mean the origin of M. puboischiofemoralis internus 2 is here reconstructed on the posterior dorsal vertebrae, similar to the crocodilian condition. The posterior dorsal vertebrae of *Thecodontosaurus* have centra with large and shallow lateral depressions [41], where the muscle might have originated. The position of the insertion site is also uncertain because of the absence of osteological correlates like the accessory trochanter in *Thecodontosaurus* and other non-theropod dinosaurs, although considering attachment sites in extant archosaurs and homologies of femoral features in the clade [49], M. puboischiofemoralis internus 2 likely inserted on the anterolateral portion of the proximal femur, anterior to the lesser trochanter. Similar to the first head of the muscle, M. puboischiofemoralis internus 2 would have primarily acted as a hip flexor in *Thecodontosaurus* [39,70].

### 3.2.8. M. flexor tibialis externus

The M. flexor tibialis externus of crocodilians is homologous to the M. flexor cruris lateralis pars pelvica of birds, both originating on the lateral surface of the postacetabular ilium and inserting on the medial side of the proximal end of the tibia via a tendon [66,71,72]. The other components of the flexor cruris group—M. puboischiotibialis and the different heads of M. flexor tibialis internus—are not reconstructed because, with the exception of M. flexor tibialis internus 3, they require a level II inference [68]. The latter is not reconstructed either because it originates on the ischium, an element that is poorly preserved in *Thecodontosaurus*.

The presence of M. flexor tibialis externus is unambiguous in non-avian dinosaurs. The ilium of *Thecodontosaurus* shows an extensive subrectangular muscle scar at the posterior part of the lateral surface of the postacetabular process (figure 5b), which likely indicates the origin site of M. flexor tibialis externus. A similar muscle scar is present in Carnian sauropodomorphs such as *Saturnalia* and *Chromogisaurus* [14,35,80]. The muscle inserted on the medial surface of the proximal tibia, likely posteriorly on the medial condyle where some faint pitting appears to be present (figure 6h). M. flexor tibialis externus functioned as a hip extensor and knee flexor [39,70].

### 3.2.9. Mm. adductor femores

Mm. adductor femores in Crocodylia and the avian homologue Mm. puboischiofemorales have two heads. The crocodilian muscle originates on the lateral side of the ischium, with different origin sites for each head, separated by the origin of M. puboischiofemoralis externus 3. Both heads insert on the posterolateral side of the distal femur, proximal to the distal condyles [66]. In birds, Mm. puboischiofemorales originate from the pubis and the ischium and insert on the posterior surface of the distal femur [62,73].

The presence of Mm. adductor femores in non-avian dinosaurs is unequivocal. Inferences about the location of the origin sites of both heads in *Thecodontosaurus* involve much uncertainty because the ischium is fragmentary. As in most early sauropodomorphs, the posterodorsal border of this element shows a distinct groove distal to the iliac peduncle (figure 5c), a feature that has been proposed as the osteological correlate for the origin of M. adductor femoris 2 in archosaurs [69]. The origin of M. adductor femoris 1 would have been located ventrally on the lateral surface of the proximal lamina, but this part of the ischium is not preserved. Both heads are inserted on the posterior aspect of the distal femur, as in extant archosaurs. Some *Thecodontosaurus* specimens show subtle pitting on a

proximodistally elongated area on the distal end of the posterolateral intermuscular line, proximal to the lateral condyle (figure 6*f*), which might represent the insertion of Mm. adductor femores. As in other bipedal non-avian dinosaurs [38,39,70], this muscle operated as a femoral adductor and retractor in *Thecodontosaurus*, also inducing some lateral rotation.

### 3.2.10. Mm. puboischiofemorales externi

This muscle complex is present across diapsids, although the number of heads varies among clades [66,68]. Three heads are found in crocodilians, two of them—M. puboischiofemoralis externus 1 and 2—originate on the medial and posterolateral surfaces of the pubis, and the third has its origin on the lateral side of the ischial blade, between the origin of the two heads of Mm. adductor femores. All three heads converge in a common insertion on the greater trochanter of the femur [66]. The avian M. obturator lateralis and medialis are homologous to puboischiofemoralis externus 1 and 2, respectively [67,78]. The origin of M. obturator lateralis is located on the lateral side of the pubis around the obturator foramen, while that of M. obturator medialis is on the medial surface of the ischium and part of the pubis, both inserting on the greater trochanter [72,73].

The presence of an Mm. puboischiofemorales externi composed of two heads is a level I inference in non-avian dinosaurs. Both heads likely had their origin on the pubis in saurischian dinosaurs [69], but this cannot be confirmed for *Thecodontosaurus* as the pubis is not preserved. Mm. puboischiofemorales externi are inserted on the greater trochanter of the femur, although this area is abraded in most *Thecodontosaurus* femora, preventing the identification of muscle scars. This muscle complex would have acted as a hip flexor and femoral rotator [39,70].

### 3.2.11. M. ischiotrochantericus

M. ischiotrochantericus is conserved across tetrapods, although its areas of attachment vary among extant archosaurs [68]. In crocodilians, this muscle originates from a large area on the medial side of the ischial blade and inserts on the lateral surface of the proximal femur, distal to the Mm. puboischiofemorales externi insertion. In birds, this muscle adopted a derived origination site on the lateral surface of the ischium and the ilioischiadic membrane and inserts on the trochanteric crest of the femur [72,73].

The presence of M. ischiotrochantericus in non-avian dinosaurs is unambiguously supported by the myology of extant archosaurs. The origin of the muscle would have been on the ischium, probably on its posterodorsal aspect. While some studies have reconstructed this attachment proximal to the iliac peduncle [31,79], others proposed a more distal origin near the M. adductor femoris 2 origin, both being separated by a ridge [35,74]. The latter could be inferred in *Thecodontosaurus*, although the former cannot be assessed because the most proximal part of the ischium is not preserved. The insertion site was on the lateral side of the proximal femur, distal to the greater trochanter. The femur of *Thecodontosaurus* show a rugosity distal to the greater trochanter and posterior to the lesser trochanter which likely marks the insertion of M. ischiotrochantericus. This muscle probably contributed to femoral retraction and lateral rotation, as in other bipedal non-avian dinosaurs [39,70].

### 3.2.12. M. caudofemoralis brevis

This is the smaller element of the caudofemoral musculature present across diapsids [68]. The M. caudofemoralis brevis of Crocodylia originates from the anterior caudal vertebrae and from the ventrolateral side of the postacetabular ilium and inserts near the fourth trochanter of the femur [55,66,71]. In birds, this muscle has its origin on the ventral surface of the postacetabular ilium and inserts on the posterior surface of the femoral shaft near the M. caudofemoralis longus insertion [62,73,81].

M. caudofemoralis brevis was unambiguously present in non-avian dinosaurs, supported by a level I inference. The origin of the muscle was in the apomorphic brevis fossa of Dinosauria, which is shallow and shows a faintly pitted surface in *Thecodontosaurus* (figure 5*b*). As in extant archosaurs, M. caudofemoralis brevis is inserted proximal to the fourth trochanter. A crescent pitted surface on the posteroproximal side of the fourth trochanter denotes the insertion of the smaller caudofemoral muscle in *Thecodontosaurus* (figure 6*e*). This pitted surface is seen in individuals of different ontogenetic stages. M. caudofemoralis brevis was one of the main femoral retractors.

### 3.2.13. M. caudofemoralis longus

M. caudofemoralis longus has a fundamental role in limb function across tetrapods [68,82]. In crocodilians, this large muscle originates on the caudal vertebrae and inserted on the femoral fourth trochanter [66,71]. In birds, M. caudofemoralis longus is reduced, originating from the pygostyle and caudal vertebrae and inserted on the posterior side of the femoral diaphysis [72,73].

A large M. caudofemoralis longus extending from the caudal vertebrae to the femur is unequivocal in non-avian dinosaurs. The caudal series of *Thecodontosaurus* is incomplete and thus the extent of the origin cannot be determined, although it probably spanned much of the anterior portion of the tail as in extant diapsids. The insertion of M. caudofemoralis longus can be located confidently on the medial side of the fourth trochanter of the femur, where a large scar in the form of a fossa with a rugose and pitted surface is present (figure 6*e*). This scar is particularly conspicuous in larger individuals of *Thecodontosaurus*. As in non-avian dinosaurs and extant diapsids [70,82], this muscle was the main hip extensor in *Thecodontosaurus*.

# 4. Discussion

## 4.1. Comparative myology and osteological correlates in dinosaurs

The appendicular myology of several non-avian dinosaurs has been reconstructed previously, and several osteological correlates have been identified. *Thecodontosaurus* shares similarities with other early branching dinosaurs and sauropodomorphs, but also shows peculiar and, in some cases, apomorphic aspects of osteological correlates that reflect muscular specializations. These unique characteristics and comparisons with previous studies are discussed below.

One of the most conspicuous osteological correlates for forelimb muscle attachment in *Thecodontosaurus* is the rugose tuberosity for the origin of the scapular head of M. triceps brachii. This feature, located on the glenoid lip of the scapula, is an autapomorphy of the taxon [41] (figure 7) because such an extensive rugosity is not seen in any other early branching sauropodomorph [13,36,45,86]. Similarly, a well-developed tubercle is also present in other early dinosaurs such as the early branching ornithischians *Eocursor* [87] and *Heterodontosaurus* [88]. In sauropods, the osteological correlate is the rugose surface of the scapular glenoid lip, which indicates a more distal origin of the muscle [26,89,90]. The elaboration of the scar in these taxa suggests a tendinous attachment of M. triceps brachii caput scapulare, as seen in extant archosaurs.

Two more osteological correlates for muscle attachment on the scapula experienced modifications throughout the evolution of Sauropodomorpha, as reviewed by Otero [26]. The scapular lateral fossa is identified as one of the sites of origin of M. supracoracoideus in dinosaurs, and this structure bounded by the acromial ridge, separating the fossa from the acromion. In early sauropodomorphs, including *Thecodontosaurus*, the acromial ridge is poorly developed, resulting in a relatively shallower scapular fossa (e.g. [36,44,63,76,86]). By contrast, the acromial ridge becomes pronounced and the scapular fossa is notably deeper in sauropods (figure 7; [26,91,92]). A greater area for the origin of M. supracoracoideus suggests that this humeral protractor may have been more developed in sauropods, reflecting an increase in the importance of shoulder flexion in quadrupedal locomotion [21]. The second osteological correlate, the ventromedial ridge of the scapula that separates the origin sites of M. subscapularis and M. scapulohumeralis posterior, is variable in sauropodomorphs. This ridge is very pronounced in early sauropodomorphs and its length along the scapular blade varies among taxa [36,44–46,86], and is later lost in sauropods (figure 7; [26]). The possible implications of this loss in the arrangement of M. subscapularis and M. scapulohumeralis posterior are unclear, although it may suggest a reduction in size of these muscles in sauropods [26].

The origin of M. brachialis on the humerus of non-avian dinosaurs is hard to determine because there are different attachment sites in extant archosaurs, whether with a crocodilian (humeral shaft) or avian (distal end of the humerus) type of origin. The lack of osteological correlates for the M. brachialis origin along the humeral shaft and the presence of the cuboid fossa on the anterior side of the distal humerus of some non-avian dinosaurs have been interpreted as indications of a distal origin of the muscle in these taxa [26,36]. Interestingly, the cuboid fossa is absent in early theropods and herrerasaurids [34] as well as early branching ornithischians [33], which has been interpreted as an indication that M. brachialis originated from the humeral shaft in these dinosaurs. By contrast, the cuboid fossa is well developed in early branching sauropodomorphs [36,44,46,76,93], and especially in

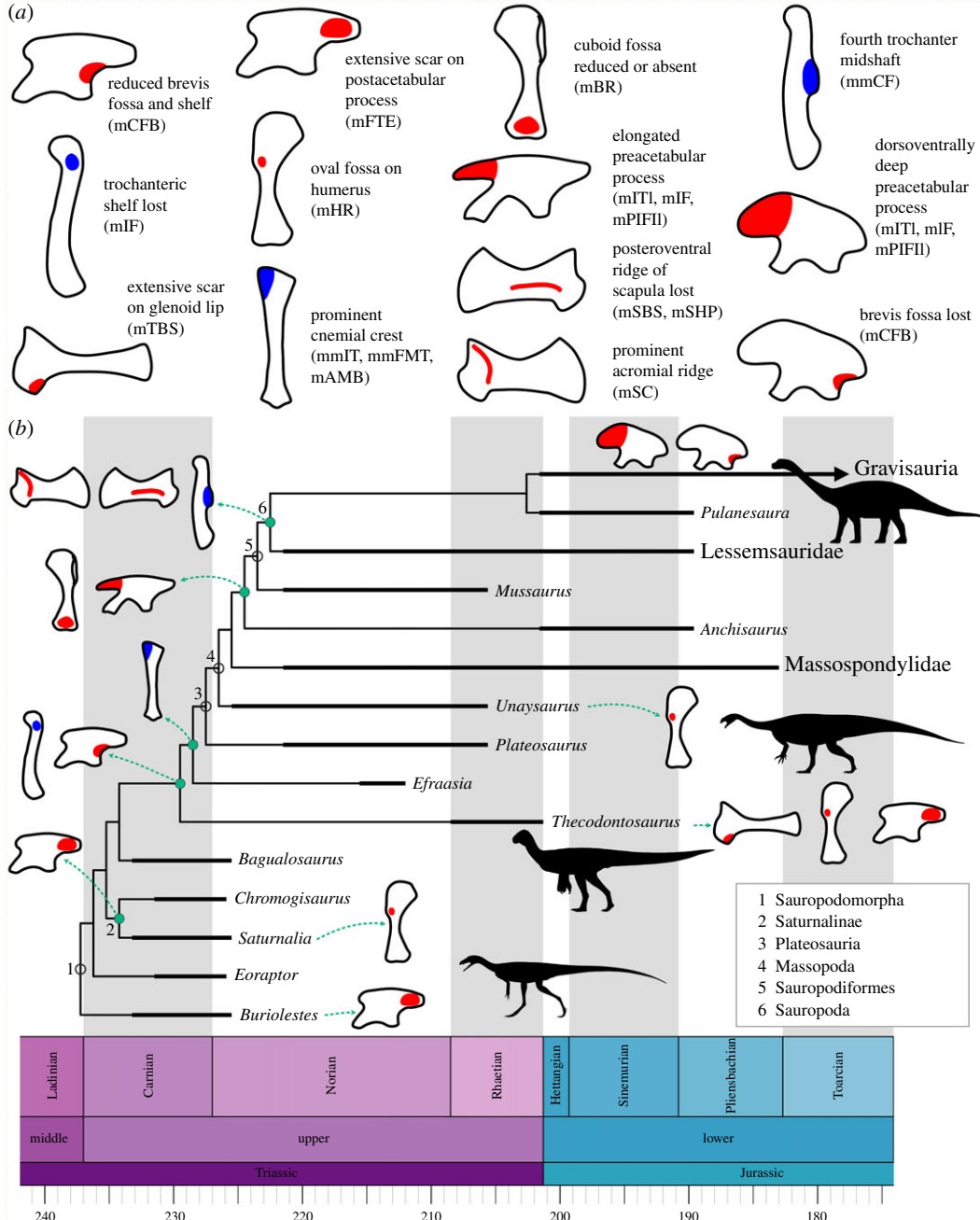

**Figure 7.** Evolution of osteological correlates of the appendicular musculature in Sauropodomorpha. Main modifications to osteological correlates associated with the appendicular musculature of sauropodomorphs. (a) Time-calibrated phylogeny of Sauropodomorpha indicating evolutionary innovations in appendicular osteological correlates of muscle attachment. (b) Simplified cladogram modified from Pol *et al.* [83], time scaled using the R packages *paleotree* [84] and *strap* [85]. *Thecodontosaurus* silhouette by Gabriel Ugueto; other sauropodomorph silhouettes taken from PhyloPic (phylopic.org): *Eoraptor* and *Plateosaurus* by Scott Hartman; *Vulcanodon* by Roberto Diaz Sibaja.

*Thecodontosaurus*, although it becomes reduced or absent in derived sauropodiforms and sauropods (figure 7; [91,94–97]). As such, the origin of M. brachialis has been inferred on the cuboid fossa in early sauropodomorphs such as *Saturnalia* and *Plateosaurus* [26,36], although the alternative humeral shaft origin has been proposed for *Mussaurus* [25]. We agree with the former avian-like distal origin and the identification of the cuboid fossa as its osteological correlate in early branching sauropodomorphs.

The presence of M. humeroradialis in non-avian dinosaurs is equivocal because there is no homologue in birds, and its reconstruction is usually complicated by the lack of clear osteological

correlates in these fossil taxa. Perhaps this muscle was present in non-avian dinosaurs, based on a tuberosity located on the lateral surface of the deltopectoral crest of some maniraptorans, resembling the origination site in crocodilians [32]. However, muscle scars or other possible osteological correlates for the M. humeroradialis origin are absent from the humeri of early theropods [34] and most early branching sauropodomorphs [26]. An exception to this is the shallow pit-like fossa on the lateral side of the deltopectoral crest in *Thecodontosaurus* and other basal sauropodomorphs such as *Saturnalia* [36], *Pampadromaeus* [45] and *Unaysaurus* [46] (figure 7). The location of this feature on the humerus, distal to the insertion site of M. latissimus dorsi and equivalent to the tuberosity seen in maniraptorans [32], suggests that this fossa represents the osteological correlate for the origin of M. humeroradialis in these taxa. These different structures may indicate a tendinous origin in maniraptorans and a fleshy attachment in sauropodomorphs.

The scar for M. flexor tibialis externus on the ilium is of phylogenetic and ontogenetic importance among early branching sauropodomorphs. As an extensive rugose area on the lateral side of the postacetabular process in Carnian sauropodomorphs, this was proposed as an apomorphy of the clade Saturnaliinae, including *Saturnalia* and *Chromogisaurus* [14,35,80]. However, this character is variable among specimens of the same species and was considered poorly diagnostic of the clade, possibly affected by ontogeny and thus absent in younger individuals [80]. *Thecodontosaurus* shares an extensive M. flexor tibialis externus scar on the postacetabular process with *Saturnalia* [35], *Chromogisaurus* [14] and *Buriolestes* [86], indicating that this scar was acquired independently in different early sauropodomorph lineages (figure 7), which complicates the use of this character as an autapomorphy of Saturnaliinae. Four fairly complete *Thecodontosaurus* ilia from two localities are known [40,41], all of similar size and bearing the rugose scar on the lateral side of the postacetabular process. Thus, *Thecodontosaurus* cannot shed light on the effect of ontogeny on this trait.

The presence of a groove on the posterodorsal edge of the proximal ischium is a ubiquitous character among early dinosaurs, seen in sauropodomorphs [35,44,93,97,98], but also in theropods [99] and ornithischians [87,100]. This peculiar feature has been interpreted as an osteological correlate for the attachment of the hindlimb musculature. This longitudinal groove could represent the origin of M. ischiotrochantericus in early dinosaurs [35,99] or, resembling the crocodilian origin of the muscle [66], the attachment site of M. adductor femoris 2 [69], an interpretation followed in subsequent dinosaur muscle reconstructions [33]. We agree with this reconstruction and consider that the ischial posterodorsal groove marks the origin of M. adductor femoris 2 in *Thecodontosaurus* and other sauropodomorphs.

The structure of the Mm. femorotibiales complex in non-avian dinosaurs is open to interpretation because extant bracketing taxa have different numbers of heads. Depending on whether the tricapitate condition of birds evolved before or after the origin of dinosaurs, the complex either had two or three heads in sauropodomorphs. Most reconstructions of the muscle in non-avian dinosaurs opt for a bicapitate architecture [30,31,33,74,79,101]. By contrast, a division of Mm. femorotibiales into three heads was suggested in *Saturnalia* based on three intermuscular lines running along the femoral shaft [35], with M. femorotibialis medialis and intermedius occupying the medial and posterior surfaces and M. femorotibialis lateralis the lateral. However, we suggest that the posteromedial intermuscular line does not mark the division of the medial head of the complex, but probably the posteromedial boundary of the muscle instead. Thus, the posterior surface of the femoral diaphysis would not be occupied by an Mm. femorotibiales attachment, being more consistent with the knee extensor role of the complex. The number and position of the intermuscular lines in this taxon are very similar to those of *Thecodontosaurus* and other early sauropodomorphs [45,77], which we consider more consistent with a bicapitate Mm. femorotibiales consisting of a medial head bounded by anterior and posteromedial intermuscular lines, and a lateral head bounded by anterior and posterolateral intermuscular lines.

The cnemial crest of the tibia is the site of insertion of the main knee extensors, included in the Triceps femoris group [31]. This structure is modest in size and has a rounded contour in Carnian sauropodomorphs [13,14,35,76], a condition that is shared by *Thecodontosaurus*. However, the cnemial crest becomes more prominent in the anteroproximal direction in later diverging sauropodomorphs such as *Efraasia* and plateosaurians (e.g. [44,65,102]; figure 7). The possible functional effect of this feature has not been tested, although it likely modified the lines of action of the main knee extensors such as Mm. iliotibiales and Mm. femorotibiales, which may have had an effect in their moment arms or in the range of knee flexion. The cnemial crest remains robust but becomes less proximally pronounced in sauropods [97,103], a modification interpreted as related to a reduced use of knee flexion and extension in quadrupedal locomotion [21].

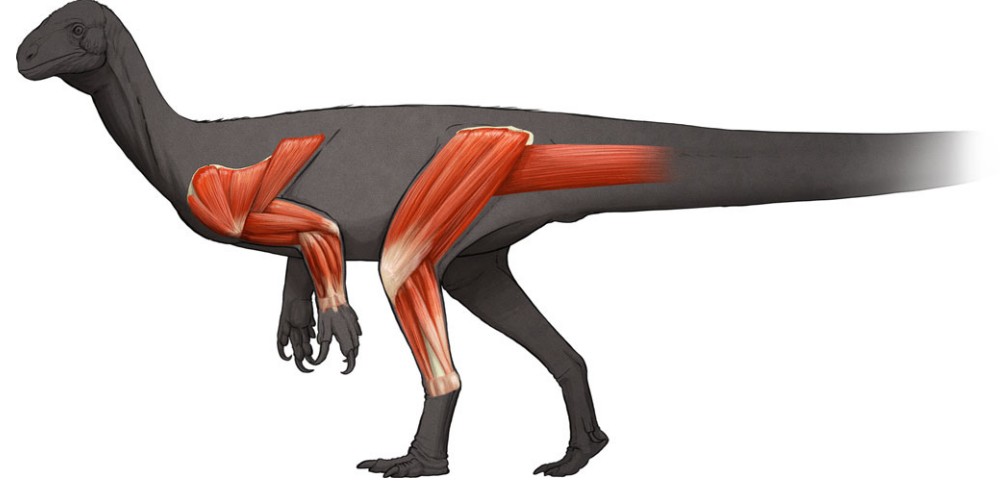

**Figure 8.** Artistic reconstruction of the limb musculature of the early diverging sauropodomorph dinosaur *Thecodontosaurus antiquus*. Artwork by Gabriel Ugueto (http://gabrielugueto.com).

## 4.2. Locomotor function in *Thecodontosaurus* and the evolution of sauropodomorph locomotion

The posture and locomotory mode of *Thecodontosaurus* have historically attracted attention, as it was one of the first dinosaurs to be studied and the first Triassic species to be named [104], and was key to the early understanding of dinosaur origins. While the first descriptions considered this taxon as a small quadruped [105], improved knowledge of dinosaur palaeobiology and comparisons with new species led to subsequent reconstruction as a biped [40]. Recent descriptions of the skeletal anatomy of *Thecodontosaurus* in comparison with other early sauropodomorphs agree with this interpretation [41], and reconstruction of its endocranial anatomy suggests it may have been an agile cursor [47]. We have shown that the appendicular musculature of *Thecodontosaurus* (figure 8) resembles that of other early branching sauropodomorphs but reveals some derived features that shed light on muscular modifications that occurred during the early evolution of the lineage.

Coupled with a suite of other forelimb anatomical modifications, reduction of the cuboid fossa of the humerus has been interpreted as a sign of reduced elbow flexion capability in sauropods associated with the acquisition of a quadrupedal and graviportal stance [95,97,106]. Minimization of this structure might represent a reduction of the attachment area and size of M. brachialis throughout the evolution of sauropodomorphs, a muscle that contributes to elbow flexion which likely lost importance with the adoption of quadrupedality. However, the deep and scarred cuboid fossa of *Thecodontosaurus* and other early branching sauropodomorphs suggests that this muscle was well developed, and these taxa retained a wide range of flexibility in the antebrachium. Similarly, the prominent scar for M. triceps brachii caput scapulare on the scapula and the marked striations on the olecranon process hint at the importance of elbow extension in *Thecodontosaurus*.

Forelimb function in non-sauropodan sauropodomorphs such as *Plateosaurus*, *Massospondylus* and *Mussaurus* has been studied by classical and computational range of motion analyses and muscle moment arms modelling [25,107–109]. These studies agree on general aspects of forelimb posture and mobility in early sauropodomorphs such as the restricted range of shoulder flexion and the limited capability for manus pronation. This evidence suggests that quadrupedality would not have been possible as the usual locomotory mode in these taxa [25,107–109]. By contrast, the earliest sauropods evolved an increased ability for humeral protraction and manus semi-pronation which predicated the postural shift in this clade [110]. The pectoral girdle and forelimb of *Thecodontosaurus* broadly resemble those of the plateosaurians *Plateosaurus*, *Massospondylus* and *Mussaurus* [44,105,111,112], although the elements are more gracile, the scapula having a long and narrow blade, the humerus showing a less developed deltopectoral crest and the zeugopodial elements being relatively more elongate [40,41]. These features suggest that *Thecodontosaurus* would have had relatively smaller areas for the attachment of the forelimb musculature compared to later-diverging taxa with larger body sizes, including its modest deltopectoral crest, insertion site of humeral protractors, or the unexpanded scapular blade, where humeral retractors originate. However, the forelimb musculature of *Thecodontosaurus* largely resembles that of other early sauropodomorphs [25,26,36,64], the differences stemming from alternative approaches in the reconstruction of certain muscles such as M. brachialis.

These similarities suggest that the forelimb of *Thecodontosaurus* would have been subject to similar mobility constraints (restricted shoulder flexion and manus pronation) to those found for other non-sauropodan sauropodomorphs [25,107–109], ruling out habitual quadrupedality for this species too. Thus, it is likely that *Thecodontosaurus* primarily used the forelimbs to grasp objects and aid food procurement, as inferred in other early sauropodomorphs [24,107]. Compared to the more gracile build of the pectoral girdle and limb skeleton of *Thecodontosaurus*, with generally smaller areas for muscle attachment, later diverging taxa evolved a more developed forelimb musculature that may have enabled more efficient manipulation [107].

The loss of the femoral trochanteric shelf in *Thecodontosaurus* has important implications for locomotor evolution of sauropodomorphs (figure 7). The origin of this structure in dinosauromorphs, incorporating the lesser trochanter, has been interpreted as an indication of the split of M. iliofemoralis into M. iliofemoralis externus and M. iliotrochantericus caudalis [78]. The trochanteric shelf is present in Carnian sauropodomorphs [14,16,35,45,76,77] but is lost in post-Carnian forms [98], and *Thecodontosaurus* is the earliest branching member of the clade without this feature (figure 7; [41]). Its loss may reflect a reversion to the plesiomorphic condition of an undivided M. iliofemoralis, thus resembling the crocodilian iliofemoral configuration [66]. The division of M. iliofemoralis externus and M. iliotrochantericus caudalis has been explained by the origin of bipedalism in dinosaurs, with these muscles acquiring the important role of stance phase hip abduction to counteract the adducting effect of the ground reaction force [78]. The loss of the trochanteric shelf might have translated into a reduced abduction ability during the stance phase by the iliofemoral musculature in post-Carnian sauropodomorphs. This might be associated with the secondary evolution of quadrupedalism in the lineage, a posture in which hip abduction is less crucial for balance because the supporting function of the forelimbs prevents the collapse of the body [78]. While most appendicular musculoskeletal features of *Thecodontosaurus* and other early-branching sauropodomorphs indicate bipedalism, the loss of the trochanteric shelf and possible reversion to an undivided M. iliofemoralis may have precluded the evolution of quadrupedalism in later larger species or might indicate some degree of facultative quadrupedalism in post-Carnian forms.

Despite this similarity with derived sauropodomorphs, most features of the iliac and femoral morphology and the associated muscles suggest bipedal posture for *Thecodontosaurus*. Different functional strategies in hip extension and flexion between putatively bipedal and quadrupedal taxa have been recovered through biomechanical analyses in the two dinosaurian lineages that evolved both postural modes: Sauropodomorpha and Ornithischia [38,39]. Elongation of the preacetabular process in Sauropodiformes and closely related taxa [92,95,113], and its further dorsoventral expansion in gravisaurians (figure 7) is related to higher moment arms of hip flexors such as M. puboischiofemoralis internus 1 and M. iliotibialis 1 [21,39]. By contrast, the short process of *Thecodontosaurus* and other early-branching sauropodomorphs would have located the lines of action of these muscles closer to the hip joint, thus reducing their moment arms and prioritizing speed, and would have left smaller areas for the origin of hip flexors. A similar functional pattern is recovered in ornithischians, with quadrupeds having greater hip flexion moment arms than bipeds for most hip flexors, mainly due to the elongation of the preacetabular process [38,114]. The relatively smaller areas for the origins of M. puboischiofemoralis internus 1 and M. iliotibialis 1 on the short preacetabular process of early sauropodomorphs may be indicative of relatively less developed hip flexors compared to those of later-branching taxa.

In sauropodomorphs, a speed versus force trade-off would also have affected the main hip extensors, the caudofemoral musculature, from the positions of their insertions. The fourth trochanter shifts to a more distal position on the femoral shaft throughout the evolution of sauropodomorphs, occupying a mid-length position in taxa close to Sauropoda [24]. The associated distal shift of the Mm. caudofemorales insertions translated into greater moment arms that compromised the speed of hip extension [82,115]. *Thecodontosaurus* resembles other early branching sauropodomorphs in the relatively proximal position of the fourth trochanter, which is consistent with fast muscle contraction and hindlimb movement. Interestingly, juvenile specimens of *Thecodontosaurus* show a more distally placed fourth trochanter, possibly negatively allometric [41], which may reflect ontogenetic variation in postural strategies, as proposed for other non-avian dinosaur species [116,117]. The brevis fossa and shelf seem to become reduced at the node comprising *Thecodontosaurus* and later diverging sauropodomorphs (figure 7), while these features are well developed in Carnian sauropodomorphs as in other early dinosaurs [13,14,16,35,45,77]. In sauropods, the brevis fossa and shelf are lost (figure 7), and thus the origin site of M. caudofemoralis brevis is usually reconstructed on the posteroventral side of the postacetabular process and the centra of the anteriormost caudal vertebrae [79,118,119].

The reduction of the brevis fossa and shelf in *Thecodontosaurus* and plateosaurians may indicate a reduced attachment of M. caudofemoralis brevis on this location and an incipient posteromedial migration of the origin of the muscle. This more posterior position of the M. caudofemoralis brevis origin likely increased the moment arm for hip extension of this muscle, in a similar fashion to the distal migration of the fourth trochanter.

## 5. Conclusion

We present a detailed description of the appendicular musculature of *Thecodontosaurus antiquus*, representing the first complete forelimb and hindlimb musculature reconstruction of an early sauropodomorph. *Thecodontosaurus* is characterized by unique muscular characters like a conspicuous muscle scar of the scapular branch of M. triceps brachii, a pit-like fossa for the origin of M. humeroradialis on the posterolateral side of the humerus and extensive scarring on the iliac blade corresponding to Mm. iliotibiales and M. flexor tibialis externus. A pronounced and pitted cuboid fossa, later reduced or lost in sauropodiforms, suggests a distal origin of M. brachialis on the humerus in early sauropodomorphs. *Thecodontosaurus* resembles Carnian sauropodomorphs in lacking the proximally pronounced cnemial crest of later diverging non-sauropodan sauropodomorphs, which is the insertion site of the main knee extensors. The brevis fossa and shelf for the origin of M. caudofemoralis brevis were reduced in *Thecodontosaurus* and later branching sauropodomorphs, which also share the loss of the trochanteric shelf, suggestive of an undivided M. iliofemoralis. Examination of juvenile specimens has not revealed important changes in osteological correlates of appendicular muscle attachment through ontogeny. The skeletal anatomy and arrangement of forelimb muscles indicate that elbow flexors and extensors were well-developed in *Thecodontosaurus*, and the range of motion at the shoulder would have been limited, suggesting that the forelimbs were not used in habitual locomotion but in manipulation, as in other early sauropodomorphs. The hindlimb musculature reconstruction suggests low moment arms for hip extensors and flexors, as expected for a bipedal, agile dinosaur. *Thecodontosaurus* also documents the phylogenetic history of important rearrangements of the appendicular musculature of sauropodomorphs, such as a reduction of the brevis fossa and shelf and the possible reversal of an undivided M. iliofemoralis which may have facilitated facultative quadrupedalism. Our myological study expands on the diversity of osteological correlates of appendicular muscles in Sauropodomorpha and the musculoskeletal modifications that occurred in this clade during the transition from small bipeds to large quadrupeds.

Data accessibility. A list of specimens examined to describe osteological correlates for muscle attachment is provided as electronic supplementary material. All specimens are catalogued and accessible in the collections of the University of Bristol Geology Department (BRSUG), Bristol, UK.

Authors' contributions. A.B.: conceptualization, data curation, formal analysis, investigation, methodology, writing-original draft, writing-review and editing; E.R.: conceptualization, supervision, writing-review and editing; M.J.B.: conceptualization, supervision, writing-review and editing.

All authors gave final approval for publication and agreed to be held accountable for the work performed therein.

Competing interests. We declare we have no competing interests.

Funding. A.B. is funded by a NERC GW4+ Doctoral Training Partnership studentship from the Natural Environment Research Council (NE/L002434/1). M.J.B. is funded by the Natural Environment Research Council BETR programme (NE/P013724/1).

Acknowledgements. We thank Claudia Hildebrandt for curation and access to the *Thecodontosaurus* material at the University of Bristol Geology Collection (BRSUG), and Simon Powell for advice on specimen photography. We are grateful to Gabriel Ugueto for producing an artistic reconstruction of the limb musculature of *Thecodontosaurus* based on our myological description. We thank Fernando Novas and an anonymous reviewer for their constructive revisions that improved the quality of our manuscript.

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
