## [Peer Review File · Royal Society Open Science]

Review History

RSOS-211356.R0 (Original submission)

Review form: Reviewer 1 (Fernando Novas)

Is the manuscript scientifically sound in its present form?

Yes

Are the interpretations and conclusions justified by the results?

Yes

Is the language acceptable?

Yes

Do you have any ethical concerns with this paper?

No

Have you any concerns about statistical analyses in this paper?

No

Recommendation?

Accept with minor revision (please list in comments)

Comments to the Author(s)

Dear authors,

I have enjoyed by reading your ms on Thecodontosaurus. In the attached pdf (see Appendix A) you will find some comments and suggestions which I believe will be useful in gaining more clarity. I congratulate the authors for the research they have conducted.

Sincerely

Fernando Novas

Review form: Reviewer 2

Is the manuscript scientifically sound in its present form?

Yes

Are the interpretations and conclusions justified by the results?

Yes

Is the language acceptable?

Yes

Do you have any ethical concerns with this paper?

No

Have you any concerns about statistical analyses in this paper?

No

Recommendation?

Major revision is needed (please make suggestions in comments)

Comments to the Author(s)

This paper is well written and represents a very useful contribution to the study of dinosaurian locomotion and paleontology as a whole. Most of my comments and corrections are relatively minor, and are indicated on the the attached pdf (see Appendix B).

The most substantial criticism I have is that the functional implications of the forelimb musculature were not very thoroughly discussed, and I think more comparisons could be made with previously published studies of sauropodomorph forelimb musculature, most notably those of Otero (Otero et al. 2017; Otero 2018). Some of the differences in this reconstruction from that of Otero are not really discussed, which leaves the reader wondering why certain choices were made. Additionally, considering the paper's aim of assessing posture/locomotion, I was expecting more comparisons with the functional analysis of *Mussaurus* mentioned above, and how *Thecodontosaurus* is similar or different. Finally, as mentioned in the comments on the pdf, I was also expecting there to be a discussion of the effects of ontogeny on the musculature/reconstruction, given that the ontogenetically broad sample was mentioned in the materials and methods. A discussion of this would be extremely valuable to the field, since so little has been published on the topic.

Decision letter (RSOS-211356.R0)

Dear Mr Ballell

The Editors assigned to your paper RSOS-211356 "Walking with early dinosaurs: appendicular myology of the Late Triassic sauropodomorph *Thecodontosaurus antiquus*" have now received comments from reviewers and would like you to revise the paper in accordance with the reviewer comments and any comments from the Editors. Please note this decision does not guarantee eventual acceptance.

Please submit your revised manuscript and required files (see below) no later than 21 days from today's (ie 29-Sep-2021) date. Note: the ScholarOne system will 'lock' if submission of the revision is attempted 21 or more days after the deadline. If you do not think you will be able to meet this deadline please contact the editorial office immediately.

on behalf of Professor Marcelo Sanchez (Associate Editor) and Kevin Padian (Subject Editor)
openscience@royalsociety.org

Associate Editor Comments to Author (Professor Marcelo Sanchez):

Associate Editor: 1

Comments to the Author:

I hope you agree the useful suggestions by the reviewers will make your study much better and you submit then the revised version.

Reviewer comments to Author

Reviewer: 1

Comments to the Author(s) (see also attached "RSOS-211356_Proof_hi - Ballell et al. Walking with early dinosaurs.pdf")

Dear authors,

I have enjoyed by reading your ms on Thecodontosaurus. In the attached pdf you will find some comments and suggestions which I believe will be useful in gaining more clarity. I congratulate the authors for the research they have conducted.

Sincerely

Fernando Novas

Reviewer: 2

Comments to the Author(s) (See also attached "RSOS-211356_Proof_hi.pdf")

This paper is well written and represents a very useful contribution to the study of dinosaurian locomotion and paleontology as a whole. Most of my comments and corrections are relatively minor, and are indicated on the attached pdf.

The most substantial criticism I have is that the functional implications of the forelimb musculature were not very thoroughly discussed, and I think more comparisons could be made with previously published studies of sauropodomorph forelimb musculature, most notably those of Otero (Otero et al. 2017; Otero 2018). Some of the differences in this reconstruction from that of Otero are not really discussed, which leaves the reader wondering why certain choices were made. Additionally, considering the paper's aim of assessing posture/locomotion, I was expecting more comparisons with the functional analysis of *Mussaurus* mentioned above, and how *Thecodontosaurus* is similar or different. Finally, as mentioned in the comments on the pdf, I was also expecting there to be a discussion of the effects of ontogeny on the musculature/reconstruction, given that the ontogenetically broad sample was mentioned in the materials and methods. A discussion of this would be extremely valuable to the field, since so little has been published on the topic.

===PREPARING YOUR MANUSCRIPT===

While not essential, it will speed up the preparation of your manuscript proof if accepted if you format your references/bibliography in Vancouver style (please see

<https://royalsociety.org/journals/authors/author-guidelines/#formatting>). You should include DOIs for as many of the references as possible.

===PREPARING YOUR REVISION IN SCHOLARONE===

Author's Response to Decision Letter for (RSOS-211356.R0)

See Appendix C.

Decision letter (RSOS-211356.R1)

Dear Mr Ballell,

It is a pleasure to accept your manuscript entitled "Walking with early dinosaurs: appendicular myology of the Late Triassic sauropodomorph *Thecodontosaurus antiquus*" in its current form for publication in Royal Society Open Science. The comments of the reviewer(s) who reviewed your manuscript are included at the foot of this letter.

The proof of your paper will be available for review using the Royal Society online proofing system and you will receive details of how to access this in the near future from our production office (openscience_proofs@royalsociety.org). We aim to maintain rapid times to publication after

acceptance of your manuscript and we would ask you to please contact both the production office and editorial office if you are likely to be away from e-mail contact to minimise delays to publication. If you are going to be away, please nominate a co-author (if available) to manage the proofing process, and ensure they are copied into your email to the journal.

on behalf of Professor Marcelo Sanchez (Associate Editor) and Kevin Padian (Subject Editor)
openscience@royalsociety.org

Editor Comments to Author:

Thanks for your revisions. Our AE comments that "The added figure 7 is excellent and the minor suggestions from the reviewers were addressed." Best wishes.

Appendix A**ROYAL SOCIETY
OPEN SCIENCE****Walking with early dinosaurs: appendicular myology of the
Late Triassic sauropodomorph *Thecodontosaurus antiquus***

Journal:	Royal Society Open Science
Manuscript ID	RSOS-211356
Article Type:	Research
Date Submitted by the Author:	19-Aug-2021
Complete List of Authors:	Ballell, Antonio; University of Bristol Rayfield, Emily; University of Bristol, School of Earth Sciences Benton, Michael; University of Bristol, Earth Sciences
Subject:	palaeontology < BIOLOGY, evolution < BIOLOGY
Keywords:	dinosaur, sauropodomorph, Triassic, muscle reconstruction, osteological correlates, locomotion
Subject Category:	Organismal and Evolutionary Biology

Author-supplied statements

Relevant information will appear here if provided.

Ethics

Does your article include research that required ethical approval or permits?:

This article does not present research with ethical considerations

Statement (if applicable):

CUST_IF_YES_ETHICS :No data available.

Data

It is a condition of publication that data, code and materials supporting your paper are made publicly available. Does your paper present new data?:

Yes

Statement (if applicable):

A list of specimens examined to describe osteological correlates for muscle attachment is provided as electronic supplementary material. All specimens are catalogued and accessible in the collections of the University of Bristol Geology Department (BRSUG), Bristol, United Kingdom.

Conflict of interest

I/We declare we have no competing interests

Statement (if applicable):

CUST_STATE_CONFLICT :No data available.

Authors' contributions

This paper has multiple authors and our individual contributions were as below

Statement (if applicable):

A.B., E.J.R. and M.J.B. conceived the study. A.B. examined and described the material. A.B. wrote the manuscript and created the figures, which were revised by all authors. All authors approved the final manuscript submission.

Walking with early dinosaurs: appendicular myology of the Late Triassic
sauropodomorph *Thecodontosaurus antiquus*

Antonio Ballell, Emily J. Rayfield and Michael J. Benton

School of Earth Sciences, University of Bristol, Life Sciences Building, Tyndall Avenue,
Bristol, BS8 1TQ, UK

**Author for correspondence:**

Antonio Ballell

e-mail: ab17506@bristol.ac.uk

**ORCID:**

Antonio Ballell: <http://orcid.org/0000-0001-8901-2398>

Emily J. Rayfield: <http://orcid.org/0000-0002-2618-750X>

Michael J. Benton: <http://orcid.org/0000-0002-4323-1824>

Abstract

Dinosaur evolution is marked by numerous independent shifts from bipedality to
quadrupedality. Sauropodomorpha is one of the lineages that transitioned from small bipedal
forms to graviportal quadrupeds, with an array of intermediate postural strategies evolving in
non-sauropodan sauropodomorphs. This locomotor shift is reflected by multiple modifications
of the appendicular skeleton, coupled with a drastic rearrangement of the limb musculature.
Here, we describe the osteological correlates and reconstruct the attachment sites of the
appendicular musculature of the Late Triassic sauropodomorph *Thecodontosaurus antiquus*,

[revised manuscript text omitted]

tibialis externus, and an undivided M. iliofemoralis. Together with the skeletal anatomy, the
arrangement of forelimb muscles indicates that elbow flexors and extensors were well-
developed, and the hindlimb musculature reconstruction suggests low moment arms for hip
extensors and flexors, as expected for a bipedal, agile dinosaur. *Thecodontosaurus* also

documents the phylogenetic history of important rearrangements of the appendicular
musculature of sauropodomorphs, such as the possible reversal of an undivided M.
iliofemoralis which may have facilitated facultative quadrupedalism. Thus, the appendicular
myology of *Thecodontosaurus* is characterized by a combination of a mostly plesiomorphic
muscle configuration with some apomorphic innovations and a few derived characters shared
with later members of Sauropodomorpha.

**Funding**

893 A. B. is funded by a NERC GW4+ Doctoral Training Partnership studentship from the Natural
Environment Research Council (NE/L002434/1). M.J.B. is funded by the Natural Environment
Research Council BETR programme (NE/P013724/1).

**Acknowledgements**

We thank Claudia Hildebrandt for curation and access to the *Thecodontosaurus* material at the
University of Bristol Geology Collection (BRSUG). We also thank Simon Powell for advice
on specimen photography.

**References**

- Sereno, P. C. 1999 The evolution of dinosaurs. *Science*. **284**, 2137–2147.
(10.1126/science.284.5423.2137)
Brusatte, S. L., Nesbitt, S. J., Irmis, R. B., Butler, R. J., Benton, M. J., Norell, M. A. 2010
The origin and early radiation of dinosaurs. *Earth-Science Reviews*. **101**, 68–100.
(10.1016/j.earscirev.2010.04.001)
Langer, M. C., Ezcurra, M. D., Bittencourt, J. S., Novas, F. E. 2010 The origin and early
evolution of dinosaurs. *Biological Reviews*. **85**, 55–110. (10.1111/j.1469-185X.2009.00094.x)

Novas, F. E. 1996 Dinosaur monophyly. *Journal of Vertebrate Paleontology*. **16**, 723–741.
(10.1080/02724634.1996.10011361)
Langer, M. C., Benton, M. J. 2006 Early dinosaurs: a phylogenetic study. *Journal of*
*Systematic Palaeontology*. **4**, 309–358. (10.1017/s1477201906001970)
Bakker, R. T., Galton, P. M. 1974 Dinosaur monophyly and a new class of vertebrates.
*Nature*. **248**, 168–172. (10.1038/248168a0)
Brusatte, S. L., Niedzwiedzki, G., Butler, R. J. 2011 Footprints pull origin and diversification
of dinosaur stem lineage deep into Early Triassic. *Proceedings of the Royal Society B*. **278**,
1107–1113. (10.1098/rspb.2010.1746)
Bates, K. T., Schachner, E. R. 2012 Disparity and convergence in bipedal archosaur
locomotion. *Journal of the Royal Society Interface*. **9**, 1339–1353. (10.1098/rsif.2011.0687)
Grinham, L. R., VanBuren, C. S., Norman, D. B. 2019 Testing for a facultative locomotor
mode in the acquisition of archosaur bipedality. *Royal Society Open Science*. **6**, 190569.
(10.1098/rsos.190569)
Demuth, O. E., Rayfield, E. J., Hutchinson, J. R. 2020 3D hindlimb joint mobility of the
stem-archosaur *Euparkeria capensis* with implications for postural evolution within
Archosauria. *Scientific Reports*. **10**, 15357. (10.1038/s41598-020-70175-y)
Charig, A. J. 1972 The evolution of the archosaur pelvis and hindlimb: an explanation in
functional terms. In *Studies in Vertebrate Evolution*. (ed. ^eds. K. A. Joysey, T. S. Kemp), pp.
121–155. Edinburgh: Oliver & Boyd.
Langer, M. C., Abdala, F., Richter, M., Benton, M. J. 1999 A sauropodomorph dinosaur
from the Upper Triassic (Carnian) of southern Brazil. *Comptes Rendus de l'Académie des*
*Sciences, Series IIA, Earth and Planetary Science*. **329**, 511–517. (10.1016/s1251-
8050(00)80025-7)

Martínez, R. N., Alcober, O. A. 2009 A basal sauropodomorph (Dinosauria: Saurischia)
from the Ischigualasto Formation (Triassic, Carnian) and the early evolution of
Sauropodomorpha. *PLoS ONE*. **4**, e4397. (10.1371/journal.pone.0004397)
Ezcurra, M. D. 2010 A new early dinosaur (Saurischia: Sauropodomorpha) from the Late
Triassic of Argentina: a reassessment of dinosaur origin and phylogeny. *Journal of Systematic*
*Palaeontology*. **8**, 371–425. (10.1080/14772019.2010.484650)
Cabreira, S. F., Schultz, C. L., Bittencourt, J. S., Soares, M. B., Fortier, D. C., Silva, L. R.,
Langer, M. C. 2011 New stem-sauropodomorph (Dinosauria, Saurischia) from the Triassic of
Brazil. *Naturwissenschaften*. **98**, 1035–1040. (10.1007/s00114-011-0858-0)
Cabreira, S. F., Kellner, A. W. A., Dias-da-Silva, S., da Silva, L. R., Bronzati, M., Marsola,
944 J. C. D., Müller, R. T., Bittencourt, J. D., Batista, B. J., Raugust, T., *et al.* 2016 A unique Late
Triassic dinosauro-morph assemblage reveals dinosaur ancestral anatomy and diet. *Current*
*Biology*. **26**, 3090–3095. (10.1016/j.cub.2016.09.040)
Martínez, R. N., Apaldetti, C., Abelin, D. 2012 Basal sauropodomorphs from the
Ischigualasto Formation. *Journal of Vertebrate Paleontology*. **32**, 51–69.
(10.1080/02724634.2013.819361)
Apaldetti, C., Martínez, R. N., Cerda, I. A., Pol, D., Alcober, O. 2018 An early trend towards
gigantism in Triassic sauropodomorph dinosaurs. *Nature Ecology & Evolution*. **2**, 1227–1232.
(10.1038/s41559-018-0599-y)
Bronzati, M., Müller, R. T., Langer, M. C. 2019 Skull remains of the dinosaur *Saturnalia*
*tupiniquim* (Late Triassic, Brazil): with comments on the early evolution of sauropodomorph
feeding behaviour. *PLoS ONE*. **14**, e0221387. (10.1371/journal.pone.0221387)
Marsola, J. C. A., Ferreira, G. S., Langer, M. C., Button, D. J., Butler, R. J. 2019 Increases
in sampling support the southern Gondwanan hypothesis for the origin of dinosaurs.
*Palaeontology*. **62**, 473–482. (10.1111/pala.12411)

Carrano, M. T. 2005 The evolution of sauropod locomotion. In *The Sauropods: evolution*
*and paleobiology*. (ed.^eds. K. A. Curry Rogers, W. J. A.), pp. 229-251. Berkeley: University
of California Press.
Benson, R. B. J., Hunt, G., Carrano, M. T., Campione, N. 2018 Cope's rule and the adaptive
landscape of dinosaur body size evolution. *Palaeontology*. **61**, 13–48. (10.1111/pala.12329)
McPhee, B. W., Benson, R. B. J., Botha-Brink, J., Bordy, E. M., Choiniere, J. N. 2018 A
giant dinosaur from the Earliest Jurassic of South Africa and the transition to quadrupedality
in early sauropodomorphs. *Current Biology*. **28**, 3143–3151. (10.1016/j.cub.2018.07.063)
Yates, A. M., Bonnan, M. F., Neveling, J., Chinsamy, A., Blackbeard, M. G. 2010 A new
transitional sauropodomorph dinosaur from the Early Jurassic of South Africa and the
evolution of sauropod feeding and quadrupedalism. *Proceedings of the Royal Society B*. **277**,
787–794. (10.1098/rspb.2009.1440)
Otero, A., Allen, V., Pol, D., Hutchinson, J. R. 2017 Forelimb muscle and joint actions in
Archosauria: insights from *Crocodylus johnstoni* (Pseudosuchia) and *Mussaurus patagonicus*
(Sauropodomorpha). *PeerJ*. **5**, e3976. (10.7717/peerj.3976)
Otero, A. 2018 Forelimb musculature and osteological correlates in Sauropodomorpha
(Dinosauria, Saurischia). *PLoS ONE*. **13**, e0198988. (10.1371/journal.pone.0198988)
Bryant, H. N., Russell, A. P. 1992 The role of phylogenetic analysis in the inference of
unpreserved attributes of extinct taxa. *Philosophical Transactions of the Royal Society of*
*London B*. **337**, 405–418. (10.1098/rstb.1992.0117)
Witmer, L. M. 1995 The extant phylogenetic bracket and the importance of reconstructing
soft tissues in fossils. In *Functional Morphology in Vertebrate Paleontology*. (ed.^eds. J. J.
Thomason), pp. 19–33. Cambridge, UK: Cambridge University Press.
Coombs, W. P. 1979 Osteology and myology of the hindlimb in the Ankylosauria (Reptilia,
Ornithischia). *Journal of Paleontology*. **53**, 666–684.

Dilkes, D. W. 2000 Appendicular myology of the hadrosaurian dinosaur *Maiasaura*
*peeblesorum* from the Late Cretaceous (Campanian) of Montana. *Transactions of the Royal*
*Society of Edinburgh: Earth Sciences*. **90**, 87–125. (10.1017/s0263593300007185)
Carrano, M. T., Hutchinson, J. R. 2002 Pelvic and hindlimb musculature of *Tyrannosaurus*
*rex* (Dinosauria: Theropoda). *Journal of Morphology*. **253**, 207–228. (10.1002/jmor.10018)
Jasinowski, S. C., Russell, A. P., Currie, P. J. 2006 An integrative phylogenetic and
extrapolatory approach to the reconstruction of dromaeosaur (Theropoda: Eumaniraptora)
shoulder musculature. *Zoological Journal of the Linnean Society*. **146**, 301–344.
(10.1111/j.1096-3642.2006.00200.x)
Maidment, S. C. R., Barrett, P. M. 2011 The locomotor musculature of basal ornithischian
dinosaurs. *Journal of Vertebrate Paleontology*. **31**, 1265–1291.
(10.1080/02724634.2011.606857)
Burch, S. H. 2014 Complete forelimb myology of the basal theropod dinosaur *Tawa hallae*
based on a novel robust muscle reconstruction method. *Journal of Anatomy*. **225**, 271–297.
(10.1111/joa.12216)
Langer, M. C. 2003 The pelvic and hind limb anatomy of the stem-sauropodomorph
*Saturnalia tupiniquim* (Late Triassic, Brazil). *PaleoBios*. **32**, 1-30.
Langer, M. C., França, M. A. G., Gabriel, S. 2007 The pectoral girdle and forelimb anatomy
of the stem-sauropodomorph *Saturnalia tupiniquim* (Upper Triassic, Brazil). *Special Papers in*
*Palaeontology*. 113–137.
Hutchinson, J. R., Anderson, F. C., Blemker, S. S., Delp, S. L. 2005 Analysis of hindlimb
muscle moment arms in *Tyrannosaurus rex* using a three-dimensional musculoskeletal
computer model: implications for stance, gait, and speed. *Paleobiology*. **31**, 676–701.

Maidment, S. C. R., Bates, K. T., Falkingham, P. L., VanBuren, C., Arbour, V., Barrett, P.
1008 M. 2014 Locomotion in ornithischian dinosaurs: an assessment using three-dimensional
computational modelling. *Biological Reviews*. **89**, 588–617. (10.1111/brv.12071)
Klinkhamer, A. J., Mallison, H., Poropat, S. F., Sinapius, G. H. K., Wroe, S. 2018 Three-
dimensional musculoskeletal modeling of the sauropodomorph hind limb: the effect of postural
change on muscle leverage. *The Anatomical Record*. **301**, 2145-2163. (10.1002/ar.23950)
Benton, M. J., Juul, L., Storrs, G. W., Galton, P. M. 2000 Anatomy and systematics of the
prosauropod dinosaur *Thecodontosaurus antiquus* from the Upper Triassic of southwest
England. *Journal of Vertebrate Paleontology*. **20**, 77–108. (10.1671/0272-
4634(2000)020[0077:aasotp]2.0.co;2)
Ballell, A., Rayfield, E. J., Benton, M. J. 2020 Osteological redescription of the Late Triassic
sauropodomorph dinosaur *Thecodontosaurus antiquus* based on new material from
Tytherington, southwestern England. *Journal of Vertebrate Paleontology*. **40**, e1770774.
(10.1080/02724634.2020.1770774)
Yates, A. M., Kitching, J. W. 2003 The earliest known sauropod dinosaur and the first steps
towards sauropod locomotion. *Proceedings of the Royal Society B*. **270**, 1753–1758.
(10.1098/rspb.2003.2417)
Upchurch, P., Barrett, P. M., Galton, P. M. 2007 A phylogenetic analysis of basal
sauropodomorph relationships: implications for the origin of sauropod dinosaurs. *Special*
*Papers in Palaeontology*. 57–90.
Otero, A., Pol, D. 2013 Postcranial anatomy and phylogenetic relationships of *Mussaurus*
*patagonicus* (Dinosauria, Sauropodomorpha). *Journal of Vertebrate Paleontology*. **33**, 1138–
1168. (10.1080/02724634.2013.769444)
Langer, M. C., McPhee, B. W., Marsola, J. C. D., Roberto-da-Silva, L., Cabreira, S. F. 2019
Anatomy of the dinosaur *Pampadromaeus barberenai* (Saurischia — Sauropodomorpha) from

the Late Triassic Santa Maria Formation of southern Brazil. *PLoS ONE*. **14**, e0212543.
(10.1371/journal.pone.0212543)
McPhee, B. W., Bittencourt, J. S., Langer, M. C., Apaldetti, C., Da Rosa, A. A. S. 2020
Reassessment of *Unaysaurus tolentinoi* (Dinosauria: Sauropodomorpha) from the Late Triassic
(early Norian) of Brazil, with a consideration of the evidence for monophyly within non-
sauropodan sauropodomorphs. *Journal of Systematic Palaeontology*. **18**, 259–293.
(10.1080/14772019.2019.1602856)
Ballell, A., King, J. L., Neenan, J. M., Rayfield, E. J., Benton, M. J. 2020 The braincase,
brain and palaeobiology of the basal sauropodomorph dinosaur *Thecodontosaurus antiquus*.
*Zoological Journal of the Linnean Society*. zlaa157.
Whiteside, D. I., Marshall, J. E. A. 2008 The age, fauna and palaeoenvironment of the Late
Triassic fissure deposits of Tytherington, South Gloucestershire, UK. *Geological Magazine*.
**145**, 105–147. (10.1017/s0016756807003925)
Hutchinson, J. R. 2001 The evolution of femoral osteology and soft tissues on the line to
extant birds (Neornithes). *Zoological Journal of the Linnean Society*. **131**, 169–197.
(10.1006/zjls.2000.0267)
Zaaf, A., Herrel, A., Aerts, P., De Vree, F. 1999 Morphology and morphometrics of the
appendicular musculature in geckoes with different locomotor habits (Lepidosauria).
*Zoomorphology*. **119**, 9–22. (10.1007/s004350050077)
Meers, M. B. 2003 Crocodylian forelimb musculature and its relevance to Archosauria.
*Anatomical Record*. **274A**, 891–916. (10.1002/ar.a.10097)
Romer, A. S. 1944 The development of tetrapod limb musculature – the shoulder region of
*Lacerta*. *Journal of Morphology*. **74**, 1–41. (10.1002/jmor.1050740102)

Abdala, V., Diogo, R. 2010 Comparative anatomy, homologies and evolution of the pectoral
and forelimb musculature of tetrapods with special attention to extant limbed amphibians and
reptiles. *Journal of Anatomy*. **217**, 536–573. (10.1111/j.1469-7580.2010.01278.x)
Fahn-Lai, P., Biewener, A. A., Pierce, S. E. 2020 Broad similarities in shoulder muscle
architecture and organization across two amniotes: implications for reconstructing non-
mammalian synapsids. *Peerj*. **8**, e8556. (10.7717/peerj.8556)
Klinkhamer, A. J., Wilhite, D. R., White, M. A., Wroe, S. 2017 Digital dissection and three-
dimensional interactive models of limb musculature in the Australian estuarine crocodile
(*Crocodylus porosus*). *PLoS ONE*. **12**, e0175079. (10.1371/journal.pone.0175079)
Howell, A. B. 1937 Morphogenesis of the shoulder architecture: Aves. *Auk*. **54**, 364–375.
Sullivan, G. E. 1962 Anatomy and embryology of the wing musculature of the domestic
fowl (*Gallus*). *Australian Journal of Zoology*. **10**, 458–518.
Jenkins, F. A., Goslow, G. E. 1983 The functional anatomy of the shoulder of the savannah
monitor lizard (*Varanus exanthematicus*). *Journal of Morphology*. **175**, 195–216.
(10.1002/jmor.1051750207)
Fearon, J. L., Varricchio, D. J. 2016 Reconstruction of the forelimb musculature of the
Cretaceous ornithomimid dinosaur *Oryctodromeus cubicularis*: implications for digging. *Journal*
*of Vertebrate Paleontology*. **36**, e1078341. (10.1080/02724634.2016.1078341)
Biewener, A. A. 2011 Muscle function in avian flight: achieving power and control.
*Philosophical Transactions of the Royal Society B*. **366**, 1496–1506. (10.1098/rstb.2010.0353)
Romer, A. S. 1922 The locomotor apparatus of certain primitive and mammal-like reptiles.
*Bulletin of the American Museum of Natural History*. **46**, 517–606.
Vanden Berge, J., Zweers, G. 1993 Myologia. In *Handbook of Avian Anatomy: Nomina*
*Anatomica Avium*. (ed. eds. J. Baumel, A. King, A. Lucas, J. Breazile, H. Evans, J. Vanden
Berge), pp. 189–247. Cambridge: Nuttall Ornithological Club.

Martínez, R. N. 2009 *Adeopapposaurus mognai*, gen. et sp. nov. (Dinosauria:
Sauropodomorpha), with comments on adaptations of basal Sauropodomorpha. *Journal of*
*Vertebrate Paleontology*. **29**, 142-164. (10.1671/039.029.0102)
Remes, K. 2008 Evolution of the pectoral girdle and forelimb in
Sauropodomorpha (Dinosauria, Saurischia): osteology, myology and function: Universität
München.
Apaldetti, C., Pol, D., Yates, A. 2013 The postcranial anatomy of *Coloradisaurus brevis*
(Dinosauria: Sauropodomorpha) from the Late Triassic of Argentina and its phylogenetic
implications. *Palaeontology*. **56**, 277–301. (10.1111/j.1475-4983.2012.01198.x)
Romer, A. S. 1923 Crocodylian pelvic muscles and their avian and reptilian homologues.
*Bulletin of the American Museum of Natural History*. **48**, 533–552.
Romer, A. S. 1927 The development of the thigh musculature of the chick. *Journal of*
*Morphology*. **43**, 347–385. (10.1002/jmor.1050430205)
Romer, A. S. 1942 The development of tetrapod limb musculature - the thigh of *Lacerta*.
*Journal of Morphology*. **71**, 251–298. (10.1002/jmor.1050710203)
Hutchinson, J. R. 2001 The evolution of pelvic osteology and soft tissues on the line to
extant birds (Neornithes). *Zoological Journal of the Linnean Society*. **131**, 123-168.
(10.1006/zjls.2000.0254)
Allen, V. R., Kilbourne, B. M., Hutchinson, J. R. 2021 The evolution of pelvic limb muscle
moment arms in bird-line archosaurs. *Science Advances*. **7**, eabe2778.
(10.1126/sciadv.abe2778)
Otero, A., Gallina, P. A., Herrera, Y. 2010 Pelvic musculature and function of *Caiman*
*latirostris*. *Herpetological Journal*. **20**, 173–184.
Hudson, G. E., Lanzillotti, P. J., Edwards, G. D. 1959 Muscles of the pelvic limb in
galliform birds. *The American Midland Naturalist*. **61**, 1–67.

Gangl, D., Weissengruber, G. E., Egerbacher, M., Forstenpointner, G. 2004 Anatomical
description of the muscles of the pelvic limb in the ostrich (*Struthio camelus*). *Anatomia,*
*Histologia, Embryologia.* **33**, 100–114. (10.1111/j.1439-0264.2003.00522.x)
Grillo, O. N., Azevedo, S. A. K. 2011 Pelvic and hind limb musculature of *Staurikosaurus*
*pricei* (Dinosauria: Saurischia). *Anais da Academia Brasileira de Ciências.* **83**, 73–98.
(10.1590/s0001-37652011000100005)
Rowe, T. 1986 Homology and evolution of the deep dorsal thigh musculature in birds and
other Reptilia. *Journal of Morphology.* **189**, 327–346. (10.1002/jmor.1051890310)
Sereno, P. C., Martínez, R. N., Alcober, O. A. 2012 Osteology of *Eoraptor lunensis*
(Dinosauria, Sauropodomorpha). *Journal of Vertebrate Paleontology.* **32**, 83-179.
(10.1080/02724634.2013.820113)
Pretto, F. A., Langer, M. C., Schultz, C. L. 2019 A new dinosaur (Saurischia:
Sauropodomorpha) from the Late Triassic of Brazil provides insights on the evolution of
sauropodomorph body plan. *Zoological Journal of the Linnean Society.* **185**, 388–416.
(10.1093/zoolinnean/zly028)
Hutchinson, J. R., Gatesy, S. M. 2000 Adductors, abductors, and the evolution of archosaur
locomotion. *Paleobiology.* **26**, 734-751. (10.1666/0094-
8373(2000)026<0734:aaateo>2.0.co;2)
Voegele, K. K., Ullmann, P. V., Lamanna, M. C., Lacovara, K. J. 2021 Myological
reconstruction of the pelvic girdle and hind limb of the giant titanosaurian sauropod dinosaur
*Dreadnoughtus schrani*. *Journal of Anatomy.* **238**, 576–597. (10.1111/joa.13334)
Garcia, M. S., Pretto, F. A., Dias-Da-Silva, S., Müller, R. T. 2019 A dinosaur ilium from
the Late Triassic of Brazil with comments on key-character supporting Saturnaliinae. *Anais da*
*Academia Brasileira de Ciências.* **91**, 19. (10.1590/0001-3765201920180614)

Hutchinson, J. R. 2002 The evolution of hindlimb tendons and muscles on the line to crown-
group birds. *Comparative Biochemistry and Physiology A*. **133**, 1051–1086. (10.1016/s1095-
6433(02)00158-7)
Gatesy, S. M. 1990 Caudofemoral musculature and the evolution of theropod locomotion.
*Paleobiology*. **16**, 170–186. (10.1017/s0094837300009866)
Müller, R. T., Langer, M. C., Bronzati, M., Pacheco, C. P., Cabreira, S. F., Dias-Da-Silva,
S. 2018 Early evolution of sauropodomorphs: anatomy and phylogenetic relationships of a
1135 remarkably well-preserved dinosaur from the Upper Triassic of southern Brazil. *Zoological*
*Journal of the Linnean Society*. **184**, 1187–1248.
Butler, R. J. 2010 The anatomy of the basal ornithischian dinosaur *Eocursor parvus* from
the lower Elliot Formation (Late Triassic) of South Africa. *Zoological Journal of the Linnean*
*Society*. **160**, 648–684. (10.1111/j.1096-3642.2009.00631.x)
Santa Luca, A. P., Crompton, A. W., Charig, A. J. 1976 A complete skeleton of the Late
Triassic ornithischian *Heterodontosaurus tucki*. *Nature*. **264**, 324–328. (10.1038/264324a0)
Borsuk-Bialynicka, M. 1977 A new camarasaurid sauropod *Opisthocoelicaudia skarzynskii*
gen. n., sp. n. from the Upper Cretaceous of Mongolia. *Palaeontologica Polonica*. **37**, 1–64.
Voegele, K. K., Ullmann, P. V., Lamanna, M. C., Lacovara, K. J. 2020 Appendicular
myological reconstruction of the forelimb of the giant titanosaurian sauropod dinosaur
*Dreadnoughtus schrani*. *Journal of Anatomy*. **237**, 133–154. (10.1111/joa.13176)
Marsh, A. D., Rowe, T. B. 2018 Anatomy and systematics of the sauropodomorph
*Sarahsaurus aurifontanalis* from the Early Jurassic Kayenta Formation. *PLoS ONE*. **13**,
e0204007. (10.1371/journal.pone.0204007)
Cooper, M. R. 1984 A reassessment of *Vulcanodon karibaensis* Raath (Dinosauria:
Saurischia) and the origin of the Sauropoda. *Palaeontologia Africana*. **25**, 203–231.

Yates, A. M. 2004 *Anchisaurus polyzelus* (Hitchcock): the smallest known sauropod
dinosaur and the evolution of gigantism among sauropodomorph dinosaurs. *Postilla*. **230**, 1–
58.
Remes, K., Ortega, F., Fierro, I., Joger, U., Kosma, R., Marín Ferrer, J. M., Ide, O. A.,
Maga, A. 2009 A new basal sauropod dinosaur from the Middle Jurassic of Niger and the early
evolution of Sauropoda. *PLoS ONE*. **4**, e6924. (10.1371/journal.pone.0006924)
Otero, A. 2010 The appendicular skeleton of *Neuquensaurus*, a Late Cretaceous saltasaurine
sauropod from Patagonia, Argentina. *Acta Palaeontologica Polonica*. **55**, 399–426.
(10.4202/app.2009.0099)
McPhee, B. W., Choiniere, J. N. 2018 The osteology of *Pulanesaura eocollum*: implications
for the inclusivity of Sauropoda (Dinosauria). *Zoological Journal of the Linnean Society*. **182**,
830–861.
Garcia, M. S., Pretto, F. A., Dias-Da-Silva, S., Müller, R. T. 2019 A dinosaur ilium from
the Late Triassic of Brazil with comments on key-character supporting Saturnaliinae. *Anais da*
*Academia Brasileira de Ciências*. **91**, 1–19. (10.1590/0001-3765201920180614)
Yates, A. M. 2003 A new species of the primitive dinosaur *Thecodontosaurus* (Saurischia:
Sauropodomorpha) and its implications for the systematics of early dinosaurs. *Journal of*
*Systematic Palaeontology*. **1**, 1–42.
Langer, M. C., Bittencourt, J. S., Schultz, C. L. 2011 A reassessment of the basal dinosaur
*Guaibasaurus candelariensis*, from the Late Triassic Caturrita Formation of south Brazil.
*Earth and Environmental Science Transactions of the Royal Society of Edinburgh*. **101**, 301–
332. (10.1017/s175569101102007x)
Sereno, P. C. 1991 *Lesothosaurus*, “fabrosaurids,” and the early evolution of Ornithischia.
*Journal of Vertebrate Paleontology*. **11**, 168–197.

Ibiricu, L. M., Martínez, R. D., Casal, G. A. 2020 The pelvic and hindlimb myology of the
basal titanosaur *Epachthosaurus sciuttoii* (Sauropoda: Titanosauria). *Historical Biology*. **32**,
773–788. (10.1080/08912963.2018.1535598)
Benton, M. J. 2012 Naming the Bristol dinosaur, *Thecodontosaurus*: politics and science in
the 1830s. *Proceedings of the Geologists Association*. **123**, 766–778.
(10.1016/j.pgeola.2012.07.012)
von Huene, F. 1932 Die fossile Reptil-Ordnung Saurischia, ihre Entwicklung und
Geschichte. *Monographien zur Geologie und Paläontologie*. **4**, 1–361.
Bonnan, M. F. 2003 The evolution of manus shape in sauropod dinosaurs: implications for
functional morphology, forelimb orientation, and phylogeny. *Journal of Vertebrate*
*Paleontology*. **23**, 595–613. (10.1671/a1108)
Klinkhamer, A. J., Mallison, H., Poropat, S. F., Sinapius, G. H. K., Wroe, S. 2018 Three-
dimensional musculoskeletal modeling of the sauropodomorph hind limb: the effect of postural
change on muscle leverage. *The Anatomical Record*. **301**, 2145–2163. (10.1002/ar.23950)
Carrano, M. T. 2005 The evolution of sauropod locomotion. In *The sauropods: evolution*
*and paleobiology*. (ed. ^eds. K. A. Curry Rogers, W. J. A.), pp. 229–251. Berkeley: University
of California Press.

[revised manuscript text omitted]

462x435mm (255 x 255 DPI)

Muscle	Abbreviation
M. latissimus dorsi	mLD
M. pectoralis	mP
M. deltoideus clavicularis	mDC
M. deltoideus scapularis	mDS
M. subscapularis	mSBS
M. subcoracoideus	mSBC
M. supracoracoideus	mSC
M. coracobrachialis brevis	mCB
M. scapulohumeralis anterior	mSHA
M. scapulohumeralis posterior	mSHP
M. triceps brachii caput scapulare	mTBS
M. triceps brachii caput lateralis	mTBL
M. triceps brachii caput	mTBM
M. biceps brachii	mBB
M. humeroradialis	mHR
M. brachialis	mBR
M. iliotibialis 1	mIT1
M. iliotibialis 2	mIT2
M. iliotibialis 3	mIT3
M. ambiens	mAMB
M. femorotibialis medialis	mFMTM
M. femorotibialis lateralis	mFMTL
M. iliofibularis	mILFB
M. iliofemoralis	mIF
M. iliofemoralis internus	mIFI
M. flexor tibialis externus	mFTE
M. adductor femoris 1	mADD1
M. adductor femoris 2	mADD2
M. puboischiofemoralis externus	mPIFE
M. ischiotrochantericus	mISTR
M. caudofemoralis brevis	mCFB
M. caudofemoralis longus	mCFL

Origin	Level of inference

Pectoral musculature

Neural spines of cervical and dorsal vertebrae	I?
Sternal plates and clavicles	I?
Lateral side of the acromion	I'
Posterolateral surface of scapular blade	I'
Medial side of scapular blade, on a depression with striations	I
Medial side of coracoid, anterior to coracoid pit	I'
Dorsolateral surface of coracoid and lateral scapular fossa	I'
Fossa on ventrolateral side of coracoid	I
Lateral surface of scapula posterodorsal to glenoid lip	I'
Ventromedial surface of scapular blade, ventral to the ventromedial ridge	I'
Tuberosity on the lateral side of the glenoid lip of the scapula	I
Posterolateral surface of the humeral shaft	I'
Posteromedial surface of the humeral shaft	I'
Anterolateral surface of the coracoid	I?
Fossa on lateral side of humerus	II
Cuboid fossa	I

Pelvic musculature

Dorsal margin of iliac preacetabular process	I
Dorsal margin of iliac body	I?
Dorsal margin of iliac postacetabular process	I
Pubic peduncle	I?
Anteromedial femoral shaft	I
Anterolateral femoral shaft	I
Lateral surface of ilium, posterior to mIF	I'
Lateral surface of ilium	I'
Preacetabular fossa of ilium	II
Lateral surface of postacetabular process	I
Lateral surface of the proximal lamina of the ischium	I?
Groove on posterodorsal side of the ischial blade	II
Pubis	II?
Posterodorsal aspect of proximal ischium	II'
Brevis fossa of ilium	II
Anterior caudal vertebrae	I?

Insertion	Level of inference
Scar on posterolateral side of humerus	I
Medial deltopectoral crest	I'
Lateral side of deltopectoral crest	I'
Posterolateral surface of proximal humerus	I'
Medial tuber of humerus	I
Medial tuber of humerus	I
Apex of deltopectoral crest	I?
Anterior surface of proximal humerus	I'
Posterior surface of proximal humerus	I'
Posterior surface of proximal humerus	I'
Olecranon process	I
Olecranon process	I
Olecranon process	I
Anteromedial side of proximal ulna and radius	I' (ulna)/ I? (radius)
Anteromedial side of proximal radius	II?
Anteromedial side of proximal ulna and radius	I' (ulna)/ I? (radius)
Cnemial crest	I
Cnemial crest	I
Cnemial crest	I
Cnemial crest	I
Cnemial crest	I
Cnemial crest	I
Anterolateral rugosity on proximal fibula	I
Lesser trochanter	II
Anteromedial proximal femur	I'
Posteromedial surface of proximal tibia	I
Scar on posterolateral distal femur	I
Scar on posterolateral distal femur	I
Greater trochanter of femur	I?
Dorsolateral trochanter of the femur	I
Proximolateral surface of fourth trochanter	I
Medial surface of fourth trochanter	I

Appendix B**ROYAL SOCIETY
OPEN SCIENCE****Walking with early dinosaurs: appendicular myology of the
Late Triassic sauropodomorph *Thecodontosaurus antiquus***

Journal:	Royal Society Open Science
Manuscript ID	RSOS-211356
Article Type:	Research
Date Submitted by the Author:	19-Aug-2021
Complete List of Authors:	Ballell, Antonio; University of Bristol Rayfield, Emily; University of Bristol, School of Earth Sciences Benton, Michael; University of Bristol, Earth Sciences
Subject:	palaeontology < BIOLOGY, evolution < BIOLOGY
Keywords:	dinosaur, sauropodomorph, Triassic, muscle reconstruction, osteological correlates, locomotion
Subject Category:	Organismal and Evolutionary Biology

Author-supplied statements

Relevant information will appear here if provided.

Ethics

Does your article include research that required ethical approval or permits?:

This article does not present research with ethical considerations

Statement (if applicable):

CUST_IF_YES_ETHICS :No data available.

Data

It is a condition of publication that data, code and materials supporting your paper are made publicly available. Does your paper present new data?:

Yes

Statement (if applicable):

A list of specimens examined to describe osteological correlates for muscle attachment is provided as electronic supplementary material. All specimens are catalogued and accessible in the collections of the University of Bristol Geology Department (BRSUG), Bristol, United Kingdom.

Conflict of interest

I/We declare we have no competing interests

Statement (if applicable):

CUST_STATE_CONFLICT :No data available.

Authors' contributions

This paper has multiple authors and our individual contributions were as below

Statement (if applicable):

A.B., E.J.R. and M.J.B. conceived the study. A.B. examined and described the material. A.B. wrote the manuscript and created the figures, which were revised by all authors. All authors approved the final manuscript submission.

Walking with early dinosaurs: appendicular myology of the Late Triassic
sauropodomorph *Thecodontosaurus antiquus*

Antonio Ballell, Emily J. Rayfield and Michael J. Benton

School of Earth Sciences, University of Bristol, Life Sciences Building, Tyndall Avenue,
Bristol, BS8 1TQ, UK

**Author for correspondence:**

Antonio Ballell

e-mail: ab17506@bristol.ac.uk

**ORCID:**

Antonio Ballell: <http://orcid.org/0000-0001-8901-2398>

Emily J. Rayfield: <http://orcid.org/0000-0002-2618-750X>

Michael J. Benton: <http://orcid.org/0000-0002-4323-1824>

**Abstract**

Dinosaur evolution is marked by numerous independent shifts from bipedality to
quadrupedality. Sauropodomorpha is one of the lineages that transitioned from small bipedal
forms to graviportal quadrupeds, with an array of intermediate postural strategies evolving in
non-sauropodan sauropodomorphs. This locomotor shift is reflected by multiple modifications
of the appendicular skeleton, coupled with a drastic rearrangement of the limb musculature.
Here, we describe the osteological correlates and reconstruct the attachment sites of the
appendicular musculature of the Late Triassic sauropodomorph *Thecodontosaurus antiquus*,

[revised manuscript text omitted]

arrangement of forelimb muscles indicates that elbow flexors and extensors were well-
developed, and the hindlimb musculature reconstruction suggests low moment arms for hip
extensors and flexors, as expected for a bipedal, agile dinosaur. *Thecodontosaurus* also

documents the phylogenetic history of important rearrangements of the appendicular
musculature of sauropodomorphs, such as the possible reversal of an undivided M.
iliofemoralis which may have facilitated facultative quadrupedalism. Thus, the appendicular
myology of *Thecodontosaurus* is characterized by a combination of a mostly plesiomorphic
muscle configuration with some apomorphic innovations and a few derived characters shared
with later members of Sauropodomorpha.

**Funding**

893 A. B. is funded by a NERC GW4+ Doctoral Training Partnership studentship from the Natural
Environment Research Council (NE/L002434/1). M.J.B. is funded by the Natural Environment
Research Council BETR programme (NE/P013724/1).

31 897 **Acknowledgements**

We thank Claudia Hildebrandt for curation and access to the *Thecodontosaurus* material at the
University of Bristol Geology Collection (BRSUG). We also thank Simon Powell for advice
on specimen photography.

**References**

- Sereno, P. C. 1999 The evolution of dinosaurs. *Science*. **284**, 2137–2147.
(10.1126/science.284.5423.2137)
Brusatte, S. L., Nesbitt, S. J., Irmis, R. B., Butler, R. J., Benton, M. J., Norell, M. A. 2010
The origin and early radiation of dinosaurs. *Earth-Science Reviews*. **101**, 68–100.
(10.1016/j.earscirev.2010.04.001)
Langer, M. C., Ezcurra, M. D., Bittencourt, J. S., Novas, F. E. 2010 The origin and early
evolution of dinosaurs. *Biological Reviews*. **85**, 55–110. (10.1111/j.1469-185X.2009.00094.x)

Novas, F. E. 1996 Dinosaur monophyly. *Journal of Vertebrate Paleontology*. **16**, 723–741.
(10.1080/02724634.1996.10011361)
Langer, M. C., Benton, M. J. 2006 Early dinosaurs: a phylogenetic study. *Journal of*
*Systematic Palaeontology*. **4**, 309–358. (10.1017/s1477201906001970)
Bakker, R. T., Galton, P. M. 1974 Dinosaur monophyly and a new class of vertebrates.
*Nature*. **248**, 168–172. (10.1038/248168a0)
Brusatte, S. L., Niedzwiedzki, G., Butler, R. J. 2011 Footprints pull origin and diversification
of dinosaur stem lineage deep into Early Triassic. *Proceedings of the Royal Society B*. **278**,
1107–1113. (10.1098/rspb.2010.1746)
Bates, K. T., Schachner, E. R. 2012 Disparity and convergence in bipedal archosaur
locomotion. *Journal of the Royal Society Interface*. **9**, 1339–1353. (10.1098/rsif.2011.0687)
Grinham, L. R., VanBuren, C. S., Norman, D. B. 2019 Testing for a facultative locomotor
mode in the acquisition of archosaur bipedality. *Royal Society Open Science*. **6**, 190569.
(10.1098/rsos.190569)
Demuth, O. E., Rayfield, E. J., Hutchinson, J. R. 2020 3D hindlimb joint mobility of the
stem-archosaur *Euparkeria capensis* with implications for postural evolution within
Archosauria. *Scientific Reports*. **10**, 15357. (10.1038/s41598-020-70175-y)
Charig, A. J. 1972 The evolution of the archosaur pelvis and hindlimb: an explanation in
functional terms. In *Studies in Vertebrate Evolution*. (ed. ^eds. K. A. Joysey, T. S. Kemp), pp.
121–155. Edinburgh: Oliver & Boyd.
Langer, M. C., Abdala, F., Richter, M., Benton, M. J. 1999 A sauropodomorph dinosaur
from the Upper Triassic (Carnian) of southern Brazil. *Comptes Rendus de l'Académie des*
*Sciences, Series IIA, Earth and Planetary Science*. **329**, 511–517. (10.1016/s1251-
8050(00)80025-7)

Martínez, R. N., Alcober, O. A. 2009 A basal sauropodomorph (Dinosauria: Saurischia)
from the Ischigualasto Formation (Triassic, Carnian) and the early evolution of
Sauropodomorpha. *PLoS ONE*. **4**, e4397. (10.1371/journal.pone.0004397)
Ezcurra, M. D. 2010 A new early dinosaur (Saurischia: Sauropodomorpha) from the Late
Triassic of Argentina: a reassessment of dinosaur origin and phylogeny. *Journal of Systematic*
*Palaeontology*. **8**, 371–425. (10.1080/14772019.2010.484650)
Cabreira, S. F., Schultz, C. L., Bittencourt, J. S., Soares, M. B., Fortier, D. C., Silva, L. R.,
Langer, M. C. 2011 New stem-sauropodomorph (Dinosauria, Saurischia) from the Triassic of
Brazil. *Naturwissenschaften*. **98**, 1035–1040. (10.1007/s00114-011-0858-0)
Cabreira, S. F., Kellner, A. W. A., Dias-da-Silva, S., da Silva, L. R., Bronzati, M., Marsola,
944 J. C. D., Müller, R. T., Bittencourt, J. D., Batista, B. J., Raugust, T., *et al.* 2016 A unique Late
Triassic dinosauro-morph assemblage reveals dinosaur ancestral anatomy and diet. *Current*
*Biology*. **26**, 3090–3095. (10.1016/j.cub.2016.09.040)
Martínez, R. N., Apaldetti, C., Abelin, D. 2012 Basal sauropodomorphs from the
Ischigualasto Formation. *Journal of Vertebrate Paleontology*. **32**, 51–69.
(10.1080/02724634.2013.819361)
Apaldetti, C., Martínez, R. N., Cerda, I. A., Pol, D., Alcober, O. 2018 An early trend towards
gigantism in Triassic sauropodomorph dinosaurs. *Nature Ecology & Evolution*. **2**, 1227–1232.
(10.1038/s41559-018-0599-y)
Bronzati, M., Müller, R. T., Langer, M. C. 2019 Skull remains of the dinosaur *Saturnalia*
*tupiniquim* (Late Triassic, Brazil): with comments on the early evolution of sauropodomorph
feeding behaviour. *PLoS ONE*. **14**, e0221387. (10.1371/journal.pone.0221387)
Marsola, J. C. A., Ferreira, G. S., Langer, M. C., Button, D. J., Butler, R. J. 2019 Increases
in sampling support the southern Gondwanan hypothesis for the origin of dinosaurs.
*Palaeontology*. **62**, 473–482. (10.1111/pala.12411)

Carrano, M. T. 2005 The evolution of sauropod locomotion. In *The Sauropods: evolution*
*and paleobiology*. (ed.^eds. K. A. Curry Rogers, W. J. A.), pp. 229-251. Berkeley: University
of California Press.
Benson, R. B. J., Hunt, G., Carrano, M. T., Campione, N. 2018 Cope's rule and the adaptive
landscape of dinosaur body size evolution. *Palaeontology*. **61**, 13–48. (10.1111/pala.12329)
McPhee, B. W., Benson, R. B. J., Botha-Brink, J., Bordy, E. M., Choiniere, J. N. 2018 A
giant dinosaur from the Earliest Jurassic of South Africa and the transition to quadrupedality
in early sauropodomorphs. *Current Biology*. **28**, 3143–3151. (10.1016/j.cub.2018.07.063)
Yates, A. M., Bonnan, M. F., Neveling, J., Chinsamy, A., Blackbeard, M. G. 2010 A new
transitional sauropodomorph dinosaur from the Early Jurassic of South Africa and the
evolution of sauropod feeding and quadrupedalism. *Proceedings of the Royal Society B*. **277**,
787–794. (10.1098/rspb.2009.1440)
Otero, A., Allen, V., Pol, D., Hutchinson, J. R. 2017 Forelimb muscle and joint actions in
Archosauria: insights from *Crocodylus johnstoni* (Pseudosuchia) and *Mussaurus patagonicus*
(Sauropodomorpha). *PeerJ*. **5**, e3976. (10.7717/peerj.3976)
Otero, A. 2018 Forelimb musculature and osteological correlates in Sauropodomorpha
(Dinosauria, Saurischia). *PLoS ONE*. **13**, e0198988. (10.1371/journal.pone.0198988)
Bryant, H. N., Russell, A. P. 1992 The role of phylogenetic analysis in the inference of
unpreserved attributes of extinct taxa. *Philosophical Transactions of the Royal Society of*
*London B*. **337**, 405–418. (10.1098/rstb.1992.0117)
Witmer, L. M. 1995 The extant phylogenetic bracket and the importance of reconstructing
soft tissues in fossils. In *Functional Morphology in Vertebrate Paleontology*. (ed.^eds. J. J.
Thomason), pp. 19–33. Cambridge, UK: Cambridge University Press.
Coombs, W. P. 1979 Osteology and myology of the hindlimb in the Ankylosauria (Reptilia,
Ornithischia). *Journal of Paleontology*. **53**, 666–684.

Dilkes, D. W. 2000 Appendicular myology of the hadrosaurian dinosaur *Maiasaura*
*peeblesorum* from the Late Cretaceous (Campanian) of Montana. *Transactions of the Royal*
*Society of Edinburgh: Earth Sciences*. **90**, 87–125. (10.1017/s0263593300007185)
Carrano, M. T., Hutchinson, J. R. 2002 Pelvic and hindlimb musculature of *Tyrannosaurus*
*rex* (Dinosauria: Theropoda). *Journal of Morphology*. **253**, 207–228. (10.1002/jmor.10018)
Jasinowski, S. C., Russell, A. P., Currie, P. J. 2006 An integrative phylogenetic and
extrapolatory approach to the reconstruction of dromaeosaur (Theropoda: Eumaniraptora)
shoulder musculature. *Zoological Journal of the Linnean Society*. **146**, 301–344.
(10.1111/j.1096-3642.2006.00200.x)
Maidment, S. C. R., Barrett, P. M. 2011 The locomotor musculature of basal ornithischian
dinosaurs. *Journal of Vertebrate Paleontology*. **31**, 1265–1291.
(10.1080/02724634.2011.606857)
Burch, S. H. 2014 Complete forelimb myology of the basal theropod dinosaur *Tawa hallae*
based on a novel robust muscle reconstruction method. *Journal of Anatomy*. **225**, 271–297.
(10.1111/joa.12216)
Langer, M. C. 2003 The pelvic and hind limb anatomy of the stem-sauropodomorph
*Saturnalia tupiniquim* (Late Triassic, Brazil). *PaleoBios*. **32**, 1-30.
Langer, M. C., França, M. A. G., Gabriel, S. 2007 The pectoral girdle and forelimb anatomy
of the stem-sauropodomorph *Saturnalia tupiniquim* (Upper Triassic, Brazil). *Special Papers in*
*Palaeontology*. 113–137.
Hutchinson, J. R., Anderson, F. C., Blemker, S. S., Delp, S. L. 2005 Analysis of hindlimb
muscle moment arms in *Tyrannosaurus rex* using a three-dimensional musculoskeletal
computer model: implications for stance, gait, and speed. *Paleobiology*. **31**, 676–701.

Maidment, S. C. R., Bates, K. T., Falkingham, P. L., VanBuren, C., Arbour, V., Barrett, P.
1008 M. 2014 Locomotion in ornithischian dinosaurs: an assessment using three-dimensional
computational modelling. *Biological Reviews*. **89**, 588–617. (10.1111/brv.12071)
Klinkhamer, A. J., Mallison, H., Poropat, S. F., Sinapius, G. H. K., Wroe, S. 2018 Three-
dimensional musculoskeletal modeling of the sauropodomorph hind limb: the effect of postural
change on muscle leverage. *The Anatomical Record*. **301**, 2145-2163. (10.1002/ar.23950)
Benton, M. J., Juul, L., Storrs, G. W., Galton, P. M. 2000 Anatomy and systematics of the
prosauropod dinosaur *Thecodontosaurus antiquus* from the Upper Triassic of southwest
England. *Journal of Vertebrate Paleontology*. **20**, 77–108. (10.1671/0272-
4634(2000)020[0077:aasotp]2.0.co;2)
Ballell, A., Rayfield, E. J., Benton, M. J. 2020 Osteological redescription of the Late Triassic
sauropodomorph dinosaur *Thecodontosaurus antiquus* based on new material from
Tytherington, southwestern England. *Journal of Vertebrate Paleontology*. **40**, e1770774.
(10.1080/02724634.2020.1770774)
Yates, A. M., Kitching, J. W. 2003 The earliest known sauropod dinosaur and the first steps
towards sauropod locomotion. *Proceedings of the Royal Society B*. **270**, 1753–1758.
(10.1098/rspb.2003.2417)
Upchurch, P., Barrett, P. M., Galton, P. M. 2007 A phylogenetic analysis of basal
sauropodomorph relationships: implications for the origin of sauropod dinosaurs. *Special*
*Papers in Palaeontology*. 57–90.
Otero, A., Pol, D. 2013 Postcranial anatomy and phylogenetic relationships of *Mussaurus*
*patagonicus* (Dinosauria, Sauropodomorpha). *Journal of Vertebrate Paleontology*. **33**, 1138–
1168. (10.1080/02724634.2013.769444)
Langer, M. C., McPhee, B. W., Marsola, J. C. D., Roberto-da-Silva, L., Cabreira, S. F. 2019
Anatomy of the dinosaur *Pampadromaeus barberenai* (Saurischia — Sauropodomorpha) from

the Late Triassic Santa Maria Formation of southern Brazil. *PLoS ONE*. **14**, e0212543.
(10.1371/journal.pone.0212543)
46 McPhee, B. W., Bittencourt, J. S., Langer, M. C., Apaldetti, C., Da Rosa, A. A. S. 2020
Reassessment of *Unaysaurus tolentinoi* (Dinosauria: Sauropodomorpha) from the Late Triassic
(early Norian) of Brazil, with a consideration of the evidence for monophyly within non-
sauropodan sauropodomorphs. *Journal of Systematic Palaeontology*. **18**, 259–293.
(10.1080/14772019.2019.1602856)
Ballell, A., King, J. L., Neenan, J. M., Rayfield, E. J., Benton, M. J. 2020 The braincase,
brain and palaeobiology of the basal sauropodomorph dinosaur *Thecodontosaurus antiquus*.
*Zoological Journal of the Linnean Society*. zlaa157.
Whiteside, D. I., Marshall, J. E. A. 2008 The age, fauna and palaeoenvironment of the Late
Triassic fissure deposits of Tytherington, South Gloucestershire, UK. *Geological Magazine*.
**145**, 105–147. (10.1017/s0016756807003925)
Hutchinson, J. R. 2001 The evolution of femoral osteology and soft tissues on the line to
extant birds (Neornithes). *Zoological Journal of the Linnean Society*. **131**, 169–197.
(10.1006/zjls.2000.0267)
Zaaf, A., Herrel, A., Aerts, P., De Vree, F. 1999 Morphology and morphometrics of the
appendicular musculature in geckoes with different locomotor habits (Lepidosauria).
*Zoomorphology*. **119**, 9–22. (10.1007/s004350050077)
Meers, M. B. 2003 Crocodylian forelimb musculature and its relevance to Archosauria.
*Anatomical Record*. **274A**, 891–916. (10.1002/ar.a.10097)
Romer, A. S. 1944 The development of tetrapod limb musculature – the shoulder region of
*Lacerta*. *Journal of Morphology*. **74**, 1–41. (10.1002/jmor.1050740102)

Abdala, V., Diogo, R. 2010 Comparative anatomy, homologies and evolution of the pectoral
and forelimb musculature of tetrapods with special attention to extant limbed amphibians and
reptiles. *Journal of Anatomy*. **217**, 536–573. (10.1111/j.1469-7580.2010.01278.x)
Fahn-Lai, P., Biewener, A. A., Pierce, S. E. 2020 Broad similarities in shoulder muscle
architecture and organization across two amniotes: implications for reconstructing non-
mammalian synapsids. *Peerj*. **8**, e8556. (10.7717/peerj.8556)
Klinkhamer, A. J., Wilhite, D. R., White, M. A., Wroe, S. 2017 Digital dissection and three-
dimensional interactive models of limb musculature in the Australian estuarine crocodile
(*Crocodylus porosus*). *PLoS ONE*. **12**, e0175079. (10.1371/journal.pone.0175079)
Howell, A. B. 1937 Morphogenesis of the shoulder architecture: Aves. *Auk*. **54**, 364–375.
Sullivan, G. E. 1962 Anatomy and embryology of the wing musculature of the domestic
fowl (*Gallus*). *Australian Journal of Zoology*. **10**, 458–518.
Jenkins, F. A., Goslow, G. E. 1983 The functional anatomy of the shoulder of the savannah
monitor lizard (*Varanus exanthematicus*). *Journal of Morphology*. **175**, 195–216.
(10.1002/jmor.1051750207)
Fearon, J. L., Varricchio, D. J. 2016 Reconstruction of the forelimb musculature of the
Cretaceous ornithomimid dinosaur *Oryctodromeus cubicularis*: implications for digging. *Journal*
*of Vertebrate Paleontology*. **36**, e1078341. (10.1080/02724634.2016.1078341)
Biewener, A. A. 2011 Muscle function in avian flight: achieving power and control.
*Philosophical Transactions of the Royal Society B*. **366**, 1496–1506. (10.1098/rstb.2010.0353)
Romer, A. S. 1922 The locomotor apparatus of certain primitive and mammal-like reptiles.
*Bulletin of the American Museum of Natural History*. **46**, 517–606.
Vanden Berge, J., Zweers, G. 1993 Myologia. In *Handbook of Avian Anatomy: Nomina*
*Anatomica Avium*. (ed. eds. J. Baumel, A. King, A. Lucas, J. Breazile, H. Evans, J. Vanden
Berge), pp. 189–247. Cambridge: Nuttall Ornithological Club.

Martínez, R. N. 2009 *Adeopapposaurus mognai*, gen. et sp. nov. (Dinosauria:
Sauropodomorpha), with comments on adaptations of basal Sauropodomorpha. *Journal of*
*Vertebrate Paleontology*. **29**, 142-164. (10.1671/039.029.0102)
Remes, K. 2008 Evolution of the pectoral girdle and forelimb in
Sauropodomorpha (Dinosauria, Saurischia): osteology, myology and function: Universität
München.
Apaldetti, C., Pol, D., Yates, A. 2013 The postcranial anatomy of *Coloradisaurus brevis*
(Dinosauria: Sauropodomorpha) from the Late Triassic of Argentina and its phylogenetic
implications. *Palaeontology*. **56**, 277–301. (10.1111/j.1475-4983.2012.01198.x)
Romer, A. S. 1923 Crocodylian pelvic muscles and their avian and reptilian homologues.
*Bulletin of the American Museum of Natural History*. **48**, 533–552.
Romer, A. S. 1927 The development of the thigh musculature of the chick. *Journal of*
*Morphology*. **43**, 347–385. (10.1002/jmor.1050430205)
Romer, A. S. 1942 The development of tetrapod limb musculature - the thigh of *Lacerta*.
*Journal of Morphology*. **71**, 251–298. (10.1002/jmor.1050710203)
Hutchinson, J. R. 2001 The evolution of pelvic osteology and soft tissues on the line to
extant birds (Neornithes). *Zoological Journal of the Linnean Society*. **131**, 123-168.
(10.1006/zjls.2000.0254)
Allen, V. R., Kilbourne, B. M., Hutchinson, J. R. 2021 The evolution of pelvic limb muscle
moment arms in bird-line archosaurs. *Science Advances*. **7**, eabe2778.
(10.1126/sciadv.abe2778)
Otero, A., Gallina, P. A., Herrera, Y. 2010 Pelvic musculature and function of *Caiman*
*latirostris*. *Herpetological Journal*. **20**, 173–184.
Hudson, G. E., Lanzillotti, P. J., Edwards, G. D. 1959 Muscles of the pelvic limb in
galliform birds. *The American Midland Naturalist*. **61**, 1–67.

Gangl, D., Weissengruber, G. E., Egerbacher, M., Forstenpointner, G. 2004 Anatomical
description of the muscles of the pelvic limb in the ostrich (*Struthio camelus*). *Anatomia,*
*Histologia, Embryologia.* **33**, 100–114. (10.1111/j.1439-0264.2003.00522.x)
Grillo, O. N., Azevedo, S. A. K. 2011 Pelvic and hind limb musculature of *Staurikosaurus*
*pricei* (Dinosauria: Saurischia). *Anais da Academia Brasileira de Ciências.* **83**, 73–98.
(10.1590/s0001-37652011000100005)
Rowe, T. 1986 Homology and evolution of the deep dorsal thigh musculature in birds and
other Reptilia. *Journal of Morphology.* **189**, 327–346. (10.1002/jmor.1051890310)
Sereno, P. C., Martínez, R. N., Alcober, O. A. 2012 Osteology of *Eoraptor lunensis*
(Dinosauria, Sauropodomorpha). *Journal of Vertebrate Paleontology.* **32**, 83-179.
(10.1080/02724634.2013.820113)
Pretto, F. A., Langer, M. C., Schultz, C. L. 2019 A new dinosaur (Saurischia:
Sauropodomorpha) from the Late Triassic of Brazil provides insights on the evolution of
sauropodomorph body plan. *Zoological Journal of the Linnean Society.* **185**, 388–416.
(10.1093/zoolinnean/zly028)
Hutchinson, J. R., Gatesy, S. M. 2000 Adductors, abductors, and the evolution of archosaur
locomotion. *Paleobiology.* **26**, 734-751. (10.1666/0094-
8373(2000)026<0734:aaateo>2.0.co;2)
Voegele, K. K., Ullmann, P. V., Lamanna, M. C., Lacovara, K. J. 2021 Myological
reconstruction of the pelvic girdle and hind limb of the giant titanosaurian sauropod dinosaur
*Dreadnoughtus schrani*. *Journal of Anatomy.* **238**, 576–597. (10.1111/joa.13334)
Garcia, M. S., Pretto, F. A., Dias-Da-Silva, S., Müller, R. T. 2019 A dinosaur ilium from
the Late Triassic of Brazil with comments on key-character supporting Saturnaliinae. *Anais da*
*Academia Brasileira de Ciências.* **91**, 19. (10.1590/0001-3765201920180614)

Hutchinson, J. R. 2002 The evolution of hindlimb tendons and muscles on the line to crown-
group birds. *Comparative Biochemistry and Physiology A*. **133**, 1051–1086. (10.1016/s1095-
6433(02)00158-7)
Gatesy, S. M. 1990 Caudofemoral musculature and the evolution of theropod locomotion.
*Paleobiology*. **16**, 170–186. (10.1017/s0094837300009866)
Müller, R. T., Langer, M. C., Bronzati, M., Pacheco, C. P., Cabreira, S. F., Dias-Da-Silva,
S. 2018 Early evolution of sauropodomorphs: anatomy and phylogenetic relationships of a
1135 remarkably well-preserved dinosaur from the Upper Triassic of southern Brazil. *Zoological*
*Journal of the Linnean Society*. **184**, 1187–1248.
Butler, R. J. 2010 The anatomy of the basal ornithischian dinosaur *Eocursor parvus* from
the lower Elliot Formation (Late Triassic) of South Africa. *Zoological Journal of the Linnean*
*Society*. **160**, 648–684. (10.1111/j.1096-3642.2009.00631.x)
Santa Luca, A. P., Crompton, A. W., Charig, A. J. 1976 A complete skeleton of the Late
Triassic ornithischian *Heterodontosaurus tucki*. *Nature*. **264**, 324–328. (10.1038/264324a0)
Borsuk-Bialynicka, M. 1977 A new camarasaurid sauropod *Opisthocoelicaudia skarzynskii*
gen. n., sp. n. from the Upper Cretaceous of Mongolia. *Palaeontologica Polonica*. **37**, 1–64.
Voegele, K. K., Ullmann, P. V., Lamanna, M. C., Lacovara, K. J. 2020 Appendicular
myological reconstruction of the forelimb of the giant titanosaurian sauropod dinosaur
*Dreadnoughtus schrani*. *Journal of Anatomy*. **237**, 133–154. (10.1111/joa.13176)
Marsh, A. D., Rowe, T. B. 2018 Anatomy and systematics of the sauropodomorph
*Sarhsaurus aurifontanalis* from the Early Jurassic Kayenta Formation. *PLoS ONE*. **13**,
e0204007. (10.1371/journal.pone.0204007)
Cooper, M. R. 1984 A reassessment of *Vulcanodon karibaensis* Raath (Dinosauria:
Saurischia) and the origin of the Sauropoda. *Palaeontologia Africana*. **25**, 203–231.

Yates, A. M. 2004 *Anchisaurus polyzelus* (Hitchcock): the smallest known sauropod
dinosaur and the evolution of gigantism among sauropodomorph dinosaurs. *Postilla*. **230**, 1–
58.
Remes, K., Ortega, F., Fierro, I., Joger, U., Kosma, R., Marín Ferrer, J. M., Ide, O. A.,
Maga, A. 2009 A new basal sauropod dinosaur from the Middle Jurassic of Niger and the early
evolution of Sauropoda. *PLoS ONE*. **4**, e6924. (10.1371/journal.pone.0006924)
Otero, A. 2010 The appendicular skeleton of *Neuquensaurus*, a Late Cretaceous saltasaurine
sauropod from Patagonia, Argentina. *Acta Palaeontologica Polonica*. **55**, 399–426.
(10.4202/app.2009.0099)
McPhee, B. W., Choiniere, J. N. 2018 The osteology of *Pulanesaura eocollum*: implications
for the inclusivity of Sauropoda (Dinosauria). *Zoological Journal of the Linnean Society*. **182**,
830–861.
Garcia, M. S., Pretto, F. A., Dias-Da-Silva, S., Müller, R. T. 2019 A dinosaur ilium from
the Late Triassic of Brazil with comments on key-character supporting Saturnaliinae. *Anais da*
*Academia Brasileira de Ciências*. **91**, 1–19. (10.1590/0001-3765201920180614)
Yates, A. M. 2003 A new species of the primitive dinosaur *Thecodontosaurus* (Saurischia:
Sauropodomorpha) and its implications for the systematics of early dinosaurs. *Journal of*
*Systematic Palaeontology*. **1**, 1–42.
Langer, M. C., Bittencourt, J. S., Schultz, C. L. 2011 A reassessment of the basal dinosaur
*Guaibasaurus candelariensis*, from the Late Triassic Caturrita Formation of south Brazil.
*Earth and Environmental Science Transactions of the Royal Society of Edinburgh*. **101**, 301–
332. (10.1017/s175569101102007x)
Sereno, P. C. 1991 *Lesothosaurus*, “fabrosaurids,” and the early evolution of Ornithischia.
*Journal of Vertebrate Paleontology*. **11**, 168–197.

Ibiricu, L. M., Martínez, R. D., Casal, G. A. 2020 The pelvic and hindlimb myology of the
basal titanosaur *Epachthosaurus sciuttoi* (Sauropoda: Titanosauria). *Historical Biology*. **32**,
773–788. (10.1080/08912963.2018.1535598)
Benton, M. J. 2012 Naming the Bristol dinosaur, *Thecodontosaurus*: politics and science in
the 1830s. *Proceedings of the Geologists Association*. **123**, 766–778.
(10.1016/j.pgeola.2012.07.012)
von Huene, F. 1932 Die fossile Reptil-Ordnung Saurischia, ihre Entwicklung und
Geschichte. *Monographien zur Geologie und Paläontologie*. **4**, 1–361.
Bonnan, M. F. 2003 The evolution of manus shape in sauropod dinosaurs: implications for
functional morphology, forelimb orientation, and phylogeny. *Journal of Vertebrate*
*Paleontology*. **23**, 595–613. (10.1671/a1108)
Klinkhamer, A. J., Mallison, H., Poropat, S. F., Sinapius, G. H. K., Wroe, S. 2018 Three-
dimensional musculoskeletal modeling of the sauropodomorph hind limb: the effect of postural
change on muscle leverage. *The Anatomical Record*. **301**, 2145–2163. (10.1002/ar.23950)
Carrano, M. T. 2005 The evolution of sauropod locomotion. In *The sauropods: evolution*
*and paleobiology*. (ed. ^eds. K. A. Curry Rogers, W. J. A.), pp. 229–251. Berkeley: University
of California Press.

[revised manuscript text omitted]

462x435mm (255 x 255 DPI)

Muscle	Abbreviation
M. latissimus dorsi	mLD
M. pectoralis	mP
M. deltoideus clavicularis	mDC
M. deltoideus scapularis	mDS
M. subscapularis	mSBS
M. subcoracoideus	mSBC
M. supracoracoideus	mSC
M. coracobrachialis brevis	mCB
M. scapulohumeralis anterior	mSHA
M. scapulohumeralis posterior	mSHP
M. triceps brachii caput scapulare	mTBS
M. triceps brachii caput lateralis	mTBL
M. triceps brachii caput	mTBM
M. biceps brachii	mBB
M. humeroradialis	mHR
M. brachialis	mBR
M. iliotibialis 1	mIT1
M. iliotibialis 2	mIT2
M. iliotibialis 3	mIT3
M. ambiens	mAMB
M. femorotibialis medialis	mFMTM
M. femorotibialis lateralis	mFMTL
M. iliofibularis	mILFB
M. iliofemoralis	mIF
M. iliofemoralis internus	mIFI
M. flexor tibialis externus	mFTE
M. adductor femoris 1	mADD1
M. adductor femoris 2	mADD2
M. puboischiofemoralis externus	mPIFE
M. ischiotrochantericus	mISTR
M. caudofemoralis brevis	mCFB
M. caudofemoralis longus	mCFL

Origin	Level of inference

Pectoral musculature

Neural spines of cervical and dorsal vertebrae	I?
Sternal plates and clavicles	I?
Lateral side of the acromion	I'
Posterolateral surface of scapular blade	I'
Medial side of scapular blade, on a depression with striations	I
Medial side of coracoid, anterior to coracoid pit	I'
Dorsolateral surface of coracoid and lateral scapular fossa	I'
Fossa on ventrolateral side of coracoid	I
Lateral surface of scapula posterodorsal to glenoid lip	I'
Ventromedial surface of scapular blade, ventral to the ventromedial ridge	I'
Tuberosity on the lateral side of the glenoid lip of the scapula	I
Posterolateral surface of the humeral shaft	I'
Posteromedial surface of the humeral shaft	I'
Anterolateral surface of the coracoid	I?
Fossa on lateral side of humerus	II
Cuboid fossa	I

Pelvic musculature

Dorsal margin of iliac preacetabular process	I
Dorsal margin of iliac body	I?
Dorsal margin of iliac postacetabular process	I
Pubic peduncle	I?
Anteromedial femoral shaft	I
Anterolateral femoral shaft	I
Lateral surface of ilium, posterior to mIF	I'
Lateral surface of ilium	I'
Preacetabular fossa of ilium	II
Lateral surface of postacetabular process	I
Lateral surface of the proximal lamina of the ischium	I?
Groove on posterodorsal side of the ischial blade	II
Pubis	II?
Posterodorsal aspect of proximal ischium	II'
Brevis fossa of ilium	II
Anterior caudal vertebrae	I?

Insertion	Level of inference
Scar on posterolateral side of humerus	I
Medial deltopectoral crest	I'
Lateral side of deltopectoral crest	I'
Posterolateral surface of proximal humerus	I'
Medial tuber of humerus	I
Medial tuber of humerus	I
Apex of deltopectoral crest	I?
Anterior surface of proximal humerus	I'
Posterior surface of proximal humerus	I'
Posterior surface of proximal humerus	I'
Olecranon process	I
Olecranon process	I
Olecranon process	I
Anteromedial side of proximal ulna and radius	I' (ulna)/ I? (radius)
Anteromedial side of proximal radius	II?
Anteromedial side of proximal ulna and radius	I' (ulna)/ I? (radius)
Cnemial crest	I
Cnemial crest	I
Cnemial crest	I
Cnemial crest	I
Cnemial crest	I
Cnemial crest	I
Anterolateral rugosity on proximal fibula	I
Lesser trochanter	II
Anteromedial proximal femur	I'
Posteromedial surface of proximal tibia	I
Scar on posterolateral distal femur	I
Scar on posterolateral distal femur	I
Greater trochanter of femur	I?
Dorsolateral trochanter of the femur	I
Proximolateral surface of fourth trochanter	I
Medial surface of fourth trochanter	I

Appendix C

Dear Mr Ballell

The Editors assigned to your paper RSOS-211356 "Walking with early dinosaurs: appendicular myology of the Late Triassic sauropodomorph *Thecodontosaurus antiquus*" have now received comments from reviewers and would like you to revise the paper in accordance with the reviewer comments and any comments from the Editors. Please note this decision does not guarantee eventual acceptance.

Many thanks. We have made all suggested changes, and we respond to all specific points below.

Please submit your revised manuscript and required files (see below) no later than 21 days from today's (ie 29-Sep-2021) date. Note: the ScholarOne system will 'lock' if submission of the revision is attempted 21 or more days after the deadline. If you do not think you will be able to meet this deadline please contact the editorial office immediately.

on behalf of Professor Marcelo Sanchez (Associate Editor) and Kevin Padian (Subject Editor)
openscience@royalsociety.org

Associate Editor Comments to Author (Professor Marcelo Sanchez):
Associate Editor: 1
Comments to the Author:

I hope you agree the useful suggestions by the reviewers will make your study much better and you submit then the revised version.

We acknowledge all changes, and provide responses to all points raised by reviewers, in this document.

Reviewer comments to Author

Reviewer: 1

Comments to the Author(s) (see also attached "RSOS-211356_Proof_hi - Ballell et al. Walking with early dinosaurs.pdf")

Dear authors,

I have enjoyed by reading your ms on Thecodontosaurus. In the attached pdf you will find some comments and suggestions which I believe will be useful in gaining more clarity. I congratulate the authors for the research they have conducted.

Sincerely

Fernando Novas

Many thanks for the kind words and the helpful comments. We have incorporated the suggestions into the manuscript.

Lines 841-842: The trochanteric shelf is also absent in neotheropods (excepting some coelophysid specimens). The point here is that same functional conclusion (i.e., reduced abduction ability tied with quadrupedalism) has to be applied to neotheropods. Is that correct?

Indeed, the trochanteric shelf is very variable among theropods and is reduced in many tetanurans, although different structures are seen in a similar position on the femur. These structures are considered character state variations of the trochanteric shelf by Hutchinson (2001), suggesting that *M. iliofemoralis externus* and *M. iliotrochantericus caudalis* remained as separate muscles throughout theropod evolution. Thus, the same functional inference proposed for post-Carnian sauropodomorphs may not be applied to neotheropods.

Hutchinson, J. R. 2001 The evolution of femoral osteology and soft tissues on the line to extant birds (Neornithes). *Zoological Journal of the Linnean Society*. 131, 169-197. (10.1006/zjls.2000.0267)

Lines 816-818: I recommend the possibility to add a new figure, consisting in a simple but informative cladogram/phylogram, indicating the sequence of evolutionary acquisitions along basal Sauropodomorpha. By doing this, the relevance of Thecodontosaurus may be more clearly explained

We have included a new figure (figure 7) that illustrates the main modifications in the appendicular osteological correlates that occurred throughout the evolution of Sauropodomorpha.

Lines 853-855: The topic of functional strategies in sauropodomorphs and ornithischians is briefly cited, but not more explanations are offered.

The similar functional pattern in hip flexion in sauropodomorphs and ornithischians have been clarified in lines 931-936.

Reviewer: 2

Comments to the Author(s) (See also attached "RSOS-211356_Proof_hi.pdf")

This paper is well written and represents a very useful contribution to the study of

dinosaurian locomotion and paleontology as a whole. Most of my comments and corrections are relatively minor, and are indicated on the attached pdf.

We thank the reviewer for their positive critiques and respond to the main issues below.

The most substantial criticism I have is that the functional implications of the forelimb musculature were not very thoroughly discussed, and I think more comparisons could be made with previously published studies of sauropodomorph forelimb musculature, most notably those of Otero (Otero et al. 2017; Otero 2018). Some of the differences in this reconstruction from that of Otero are not really discussed, which leaves the reader wondering why certain choices were made. Additionally, considering the paper's aim of assessing posture/locomotion, I was expecting more comparisons with the functional analysis of *Mussaurus* mentioned above, and how *Thecodontosaurus* is similar or different.

We have included more references to Otero et al. (2017) and Otero (2018) throughout the Results and Discussion sections, especially in the points where our reconstruction differed from those. We have also added a paragraph in the Discussion comparing forelimb function between *Thecodontosaurus* and *Mussaurus*, as well as other early sauropodomorphs.

Finally, as mentioned in the comments on the pdf, I was also expecting there to be a discussion of the effects of ontogeny on the musculature/reconstruction, given that the ontogenetically broad sample was mentioned in the materials and methods. A discussion of this would be extremely valuable to the field, since so little has been published on the topic.

Lines 868-869: You don't really go into the ontogeny stuff very much, even though at the beginning you note you had a pretty wide ontogenetic sample. It left me curious about any other possible ontogenetic signals in the musculature, be they shifts in attachments or the relative development of osteological correlates. This would be a potentially valuable contribution since almost no reconstructions have dealt with the effects of ontogeny before.

The reviewer raises an important point. While the *Thecodontosaurus* material is abundant and composed of specimens from different presumed ontogenetic stages, many of these are fragmentary or do not represent elements of the girdle and limbs. We have included a list of examined specimens as supplementary material (Table S1), where we indicate those that are presumed to be juveniles or subadults. Unfortunately, we have not found major differences in the presence or extent of the osteological correlates discussed here from these specimens, with the only exception of the position of the fourth trochanter. This is the reason why we do not extend on the issue of ontogenetic variation in the Discussion.